# Guidance landscapes unveiled by quantitative proteomics to control reinnervation in adult visual system

Noemie Vilallongue [1,3], Julia Schaeffer[1,3], Anne-Marie Hesse[2], Céline Delpech [1], Béatrice Blot[1], Antoine Paccard [1], Elise Plissonnier[1], Blandine Excoffier[1], Yohann Couté [2], Stephane Belin[1,4] & Homaira Nawabi [1,4] ✉

In the injured adult central nervous system (CNS), activation of pro-growth molecular pathways in neurons leads to long-distance regeneration. However, most regenerative fibers display guidance defects, which prevent reinnervation and functional recovery. Therefore, the molecular characterization of the proper target regions of regenerative axons is essential to uncover the modalities of adult reinnervation. In this study, we use mass spectrometry (MS)-based quantitative proteomics to address the proteomes of major nuclei of the adult visual system. These analyses reveal that guidance-associated molecules are expressed in adult visual targets. Moreover, we show that bilateral optic nerve injury modulates the expression of specific proteins. In contrast, the expression of guidance molecules remains steady. Finally, we show that regenerative axons are able to respond to guidance cues ex vivo, suggesting that these molecules possibly interfere with brain target reinnervation in adult. Using a long-distance regeneration model, we further demonstrate that the silencing of specific guidance signaling leads to rerouting of regenerative axons in vivo. Altogether, our results suggest ways to modulate axon guidance of regenerative neurons to achieve circuit repair in adult.

In adult mammals, neurons from the central nervous system (CNS) are unable to regenerate spontaneously following injury, leading to permanent and irreversible cognitive and motor disabilities in patients. This failure of regeneration has two main components: the inhibitory environment of the lesion site and the intrinsic properties of neurons themselves[1]. Indeed, over the course of development, CNS neurons progressively lose their regenerative capabilities[2], and this decline is exacerbated after axon injury[3]. Recent studies combining genetic manipulations and pharmacological approaches have led to robust CNS regeneration, in particular in the visual system, one of the best studied CNS models in the field. In the mouse visual system, axons are

able to grow over long distances: up to several centimeters from the eye ball to the brain[3–5]. Yet, despite long-distance growth, regenerative axons fail to resume their original path and grow in a disorganized manner with numerous guidance defects[6,7], preventing functional recovery. Indeed, regenerative retinal ganglion cells (RGC) axons display strong guidance defects[8,9] as they get lost in the optic chiasm or grow in the contralateral optic nerve[3,10]. These misguided axons may jeopardize functional recovery, with several issues arising: the number of regenerative axons reaching their proper targets may be insufficient to reform a functional circuit and/or lost axons may form aberrant connections, impairing further the formation of a functioning

[1]University Grenoble Alpes, Inserm, U1216, Grenoble Institut Neurosciences, 38000 Grenoble, France. [2]University Grenoble Alpes, INSERM, CEA, UMR BioSanté U1292, CNRS, CEA, FR2048, 38000 Grenoble, France. [3]These authors contributed equally: Noemie Vilallongue, Julia Schaeffer. [4]These authors jointly supervised this work: Stephane Belin, Homaira Nawabi. ✉e-mail: homaira.nawabi@inserm.fr

neuronal circuit. Therefore, a critical question remains: can regenerative axons be guided in a mature brain?

During embryonic development, thousands of neurons project their axons over long distances to reach their functional targets. Axon navigation is oriented by neuronal response to guidance cues at critical choice points[11,12] such as the optic chiasm. Guidance molecules, either attractive or repulsive, comprise soluble molecules that can act over long distances, such as secreted Semaphorins, Slits and Netrins, and transmembrane proteins acting locally, such as transmembrane Semaphorins and Ephrins. It is now well accepted that axon trajectory is not defined by a single ligand/receptor interaction. Indeed, it involves crosstalk among several cues or receptors that are specific of the axonal path[13–15]. Guidance molecules act in addition to other factors such as the molecular machinery specific to each neuronal population[16], the substrate stiffness[17] or the neuronal activity[18,19]. Furthermore, over the past years several molecules have been described to be involved in axon guidance such as morphogens (Shh, Wnt)[20,21], growth factors (TGFβ, BMP)[22] and adhesion molecules (L1CAM, NrCAM)[23]. The expression pattern and guidance activity of these cues have been studied during embryogenesis, allowing to generate a comprehensive guidance map during development. Notably, their role during midline crossing has been largely characterized in the developing visual system[24].

In contrast, in the context of regeneration in adult CNS, many unknowns of axon guidance remain, including whether regenerating axons have a functional machinery to respond to external guidance cues, or if the integration of these signalings could influence pathfinding and connectivity of regenerating axons. Thus, a characterization of the mature neuronal environment is essential to properly guide regenerative axons and obtain functional recovery. So far, most studies have focused on the lesion site itself and considered guidance cues as inhibitory for axon growth[25,26]. However, with long-distance CNS regeneration models, axons overtake this lesion site. Therefore, it is critical to consider guidance cues for their primary role in axon pathfinding in order to overcome the failure of reinnervation. So far there is little data regarding guidance cues expression and their function in mature brain.

Here, we used mass spectrometry (MS)-based proteomics to analyze the protein content of the major functional nuclei in the mouse adult visual system innervated by RGC, as well as the optic chiasm, a critical choice point, where many guidance defects are observed during regeneration[3,7,10]. This way, we established the guidance landscape of mature visual system. Our results show that many guidance and guidance-associated factors (adhesion molecules, extracellular matrix components) are still expressed in the mature brain. We found that the expression of guidance cues and receptors remains stable upon injury, suggesting that the adult brain has an intrinsic guidance signature that may affect navigation of regenerating axons and more generally the connectivity of the injured circuit. Finally, using ex vivo and in vivo approaches, we demonstrate that (i) regenerative axons are still able to respond to guidance cues in the adult CNS and (ii) silencing of specific guidance signaling results in the reorientation of the trajectory of regenerative axons at the critical choice point. Indeed, the modulation of Ephrin-B3 or Sema4D signaling controls optic chiasm crossing. Altogether, our study provides evidence that axon guidance is still functional in the mature CNS and is particularly relevant in the context of axon regeneration and functional recovery.

## Results

### MS-based proteomic characterization of visual targets in the adult brain

The visual system has been thoroughly studied during development[27]. Indeed, retinal ganglion cells (RGC) projections from the retina to the brain via the optic nerves are well characterized and guidance molecules that control this patterning have been identified, in particular for the process of midline crossing at the optic chiasm[27]. Regarding the functional targets, several recent large scale studies have unraveled gene expression regulation during development and in adult[28–30], in particular in the developing lateral geniculate nucleus (LGN)[31,32]. Here, we used MS-based proteomics to characterize the protein composition of mature brain regions innervated by RGC axons. To this end, we microdissected the selected brain regions (labelled with Alexa-coupled cholera toxin B injected into the eye): the suprachiasmatic nucleus (SCN), the ventral and the dorsal lateral geniculate nuclei (dLGN and vLGN) and the superior colliculus (SCol) (Fig. 1a–c). Since the optic chiasm is a critical intermediate region in which the majority of RGC axons get lost during regeneration, we also microdissected the adult optic chiasm to decipher cues that may influence guidance in adult (Fig. 1b, c). Thus, we obtained the proteomes of adult visual targets as presented in Supplementary Data 1.

Focusing on the 3000 most abundant proteins from each brain region - as ranked according to the intensity-based absolute

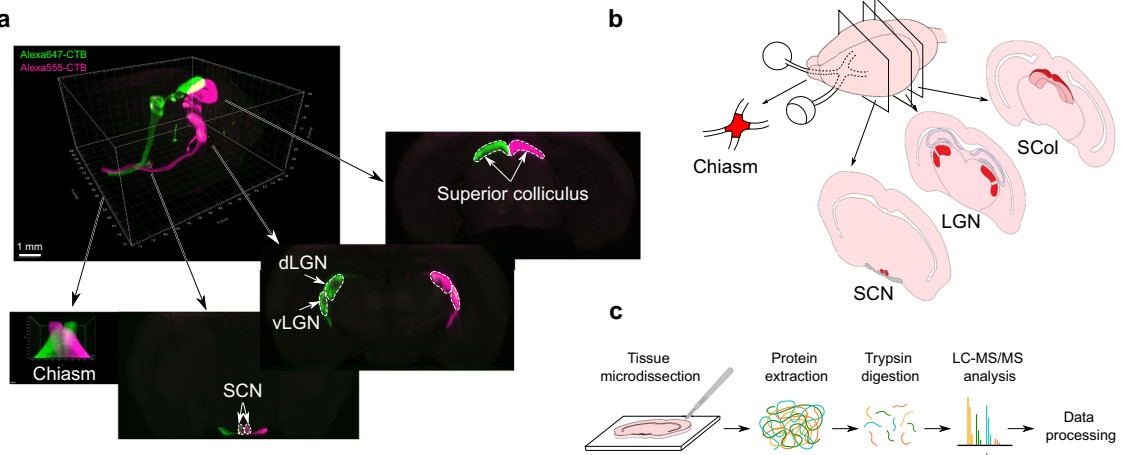

**Fig. 1 | Proteomic analysis of adult visual targets. a** 3D cleared brain imaged using light sheet microscopy highlighting the regions of interest. The two eyes were injected with Alexa555-conjugated CTB and Alexa647-conjugated CTB to allow visualization of ipsi- and contralateral projections within each visual brain target. **b** Diagram of the mouse visual targets used for the study. Primary targets of RGC axons (indicated in red) and the optic chiasm were collected for the MS analysis. **c** Experimental design. Proteins were extracted from microdissected brain regions and analysed by LC-MS/MS after trypsin digestion. LC-MS/MS, liquid chromatography coupled to tandem mass spectrometry; SCN, suprachiasmatic nucleus; LGN, lateral geniculate nucleus; SCol superior colliculus. Scalebar: 1 mm.

quantification (iBAQ values[33]), gene ontology (GO) analysis conducted with DAVID[34] revealed a high enrichment in proteins characteristic of neuronal activity and in proteins reflecting a high connectivity between the neuronal and glia partners in the different brain targets (Supplementary Fig. 1).

## Adult visual targets express guidance and guidance-associated factors

We next aimed to characterize the molecular environment of adult axons to understand whether it could affect their guidance potential in a regenerative context. For that, we focused on guidance factors, as well as on proteins whose function or expression is linked to adhesion and extracellular matrix (Supplementary Data 2, Fig. 2, Supplementary Fig. 2a–d). Analysis of the proteomes in all visual targets revealed expression of classical guidance ligands and receptors of the Semaphorin, Slit, Ephrin, Netrin and cell adhesion molecule (CAM) families (Fig. 2a), mostly described for their guidance role of navigating axons during development. Besides, some guidance molecules play a physiological role in the adult CNS, such as synapse functioning[35–37]. Very interestingly, our datasets revealed the presence of 11 guidance ligands and 13 associated receptors (thereafter designated as "guidance factors") in primary visual targets of the intact adult brain. (Fig. 2a, b).

Among the 12 proteins detected in all targets, some are permissive to axon growth and guidance such as the immunoglobulin cell adhesion molecules NCAM1 known to induce neurite growth[23], and NCAM2 recently shown to be required for growth cone formation[38,39]. Using immunofluorescence on mature brain sections, we found that NCAM1 is mostly expressed around GFAP + astrocytes and NeuN+ neurons (Fig. 2c, d, d') depending on the brain region. Other examples of adhesion proteins include members of the synaptic cell adhesion molecule family CADM1, CADM2, CADM3 and CADM4 were also identified in all targets (Supplementary Fig. 2b). These cell adhesion molecules were demonstrated to be involved in axon pathfinding, in particular for midline crossing during development[40].

Furthermore, we found molecules that are repulsive, such as Ephrin-B3 (Efnb3) detected in all visual targets (Fig. 2a, e, e'). Ephrin-B3 has been described for its role as a repulsive midline barrier for many ipsilateral axons in the developing spinal cord[41]. Interestingly, some guidance ligands were detected only in specific targets. This is the case for the repulsive guidance cue Sema4D in the chiasm, the dLGN and the SCol (Fig. 2a). We verified this result by immunofluorescence and found that Sema4D is highly expressed in the chiasm, accumulated by Olig2+ oligodendrocytes, whereas the SCN does not express it (Fig. 2c, f, f'). In addition, we found expression of Ephrin-B3, Sema4D and Sema7A associated with NeuN+ cells in the LGN (Fig. 2e', f', g'), which is the case of many guidance cues in the brain targets (Fig. 2c).

In our study, we detected 55 proteins of the extracellular matrix (ECM) (Supplementary Data 2 and 3, Supplementary Fig. 2c), including collagens and proteoglycans. For example, we found expression of CSPG4, which inhibits axon growth in vitro[42], although it is yet unclear whether NG2-expressing cells, a major cellular component of the glial scar, provides in fact a stabilized substrate to support axon growth after spinal cord injury[43]. We validated CSPG4 expression in the dLGN and vLGN, where CSPG4 accumulates in the extra-neuronal space (Fig. 2c, h, h'). Together, our datasets highlight the rich ECM environment of the target regions of RGC axons, which is of particular relevance to a context of axon growth and guidance.

Finally, we analysed proteins related to axonogenesis, axon extension, cytoskeleton, growth cone and axon guidance, a subset of proteins that we termed axon growth and guidance. We found expression of 112 proteins, including canonical guidance receptors of the Eph, Neuropilin, Plexin and Robo families (Supplementary Data 3, Fig. 2b, Supplementary Fig. 2d). We also found the protein doublecortin-like kinase 2 (DCLK2) expressed in NeuN+ cells of all visual targets (Supplementary Fig. 2e, f). DCLK2 was recently shown to promote axon growth via induction of growth cone reformation[44], an essential step for the axon to interact with its environment and to respond to guidance cues.

## Optic nerve injury modulates the proteome of the visual targets

In order to understand the influence of the injury on the proteomes of visual targets, we performed bilateral optic nerve crush in adult (6 week-old) mice (Fig. 3a). This lesion leads to a full degeneration of RGC axons (Fig. 3b), resulting in loss of connection between the eye and the visual targets. We collected the brain targets four weeks later, at a time point when regenerative axons potentially reach the visual target regions in long-distance regeneration models[3,4]. Biological replicates of the injured condition showed high consistency, as shown by the scatterplots of protein abundances (Supplementary Fig. 3). For each brain target, principal component analysis (PCA) showed clustering of replicates mainly depending on the condition (Supplementary Fig. 4a–e).

Very interestingly, injury leads to modification of the visual targets' proteomes (Supplementary Data 4, Fig. 3c–g, Supplementary Fig. 4f-j). While most studies have highlighted injury-induced changes in lesioned neurons themselves or at the lesion site, here we unravel differences in protein expression in targets that are anatomically far from the injury site. Focusing on the chiasm, our datasets reveal a high number of differentially expressed proteins (311), among which 165 are upregulated and 146 are downregulated after the injury (Supplementary Data 4, Fig. 3c, Supplementary Fig. 4f). Interestingly, proteins upregulated after injury are associated with cell adhesion and extracellular matrix (eg upregulation of Galectin-3 (Lgals3) and of Fibromodulin (Fmod)) (Fig. 3c, h), supporting the hypothesis of a remodeling of the extracellular environment of the intermediate target after injury. On the other hand, proteins downregulated include Synaptotagmin-2 (Syt2) and Exportin-T (Xpot), which are associated with synapse and transport (Fig. 3c, h). This finding is consistent with the alteration of the axonal features 28 days post-crush.

In the visual targets (SCN, vLGN, dLGN and SCol), we found smaller numbers of differentially expressed proteins (Fig. 3d–g, Supplementary Fig. 4g–j). DAVID analysis allowed us to highlight a robust enrichment of terms related to inflammatory response and to collagen in the list of proteins upregulated in the vLGN (Fig. 4e, h). The local changes in inflammatory proteins observed in these distal targets (upregulation of Complement C1q subcomponents (C1QA, C1QB, C1QC)) may be a consequence of RGC axon degeneration (glial activation, debris clearing) or may reflect the protein remodeling of the targets themselves in response to the injury. Conversely, we found a downregulation of structural components of the axon, for example the microtubule-associated protein tau (MAPT) involved in the stability of microtubules in the axon (Fig. 4e, h).

Interestingly, our proteomic analysis also revealed that the regulation of protein expression is dependent on the target itself. For example, we found that the ECM glycoprotein Tenascin-C is unchanged in the chiasm after optic nerve injury, while downregulated in the SCN, as confirmed by Western blot analysis (Fig. 3i). Tenascin-C displays bidirectional activity during development, depending on neuronal subtypes and integrin receptor expression[45]. Its upregulation in the glial scar following CNS injury seems to be associated with a growth-promoting, integrin-dependent activity[46,47]. Conversely, we found an upregulation of GFAP in the chiasm after optic nerve injury, while its expression is not changed in the SCN, as confirmed by immunofluorescence (Fig. 3j). GFAP is a marker of reactive astrocytes that reflects pathological conditions[48]. Altogether, these results suggest that brain targets respond to axon injury even at a distance from the lesion. These modifications reveal that the adult brain is a dynamic environment and that there is some specificity among the different brain regions. These changes influence the capacity of regenerative

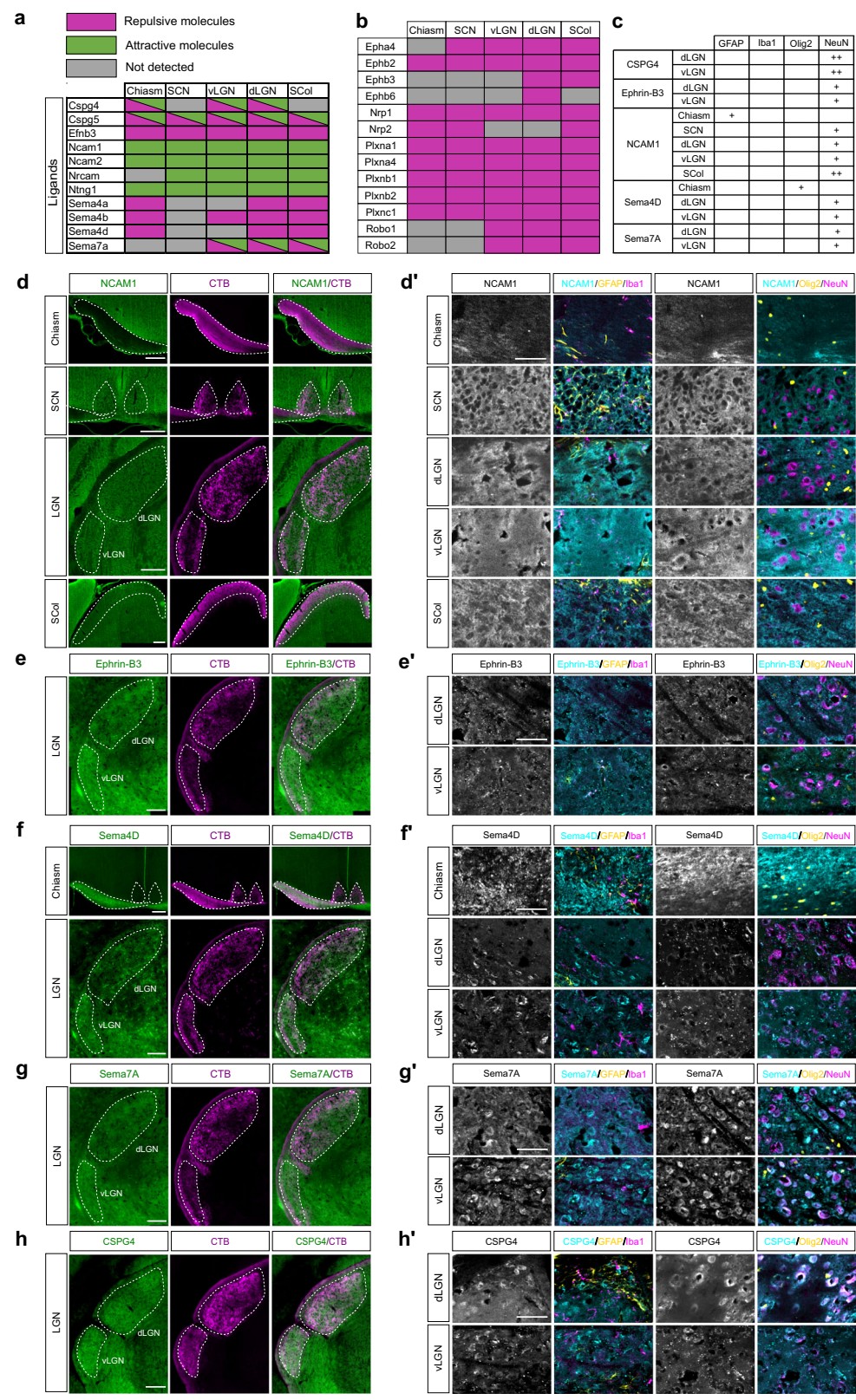

axons to resume developmental processes to reconnect their proper target.

## Expression of guidance molecules remains steady in the brain after injury but is developmentally regulated

In most models of regeneration, only few axons are observed to enter the brain targets[4]. Our hypothesis is that the presence of guidance molecules in the adult visual system interferes with the navigation of regenerating axons, preventing them to reconnect their proper brain target. While our data show that injury modulates the expression of several proteins even in distal targets of the visual system, it reveals no difference in expression level of guidance factors (Fig. 4a–e). These results were confirmed by Western Blot analysis for several examples in all target regions (Fig. 4f–j). They show that the optic nerve injury

**Fig. 2 | Validation of proteomic analysis. a** Table representing the detection (green for attractive molecules, magenta for repulsive molecules) or the absence of detection (grey) of guidance ligands in each brain target. **b** Table representing the detection (green for attractive molecules, magenta for repulsive molecules) or the absence of detection (grey) of guidance receptors in each brain target. **c** Table summarizing the type of cells expressing the guidance molecules of interest. **d** Epifluorescence images of NCAM1 immunofluorescent labelling in the adult chiasm, SCN, LGN and SCol (co-labelled with CTB). **d'** Confocal images of NCAM1 and different cell populations markers (GFAP, Iba1, Olig2, NeuN) in the adult intact chiasm, SCN, dLGN, vLGN and SCol. **e** Epifluorescence images of Ephrin-B3 immunofluorescent labelling in the adult LGN (co-labelled with CTB). **e'** Confocal images of Ephrin-B3 and different cell populations markers (GFAP, Iba1, Olig2, NeuN) in the adult intact dLGN and vLGN. **f** Epifluorescence images of Sema4D immunofluorescent labelling in the adult chiasm and LGN (co-labelled with CTB). **f'** Confocal images of Sema4D and different cell populations markers (GFAP, Iba1, Olig2, NeuN) in the adult intact chiasm, dLGN and vLGN. **g** Epifluorescence images of Sema7A immunofluorescent labelling in the adult intact LGN (co-labelled with CTB). **g'** Confocal images of Sema7A and different cell populations markers (GFAP, Iba1, Olig2 and NeuN) in the adult intact dLGN and vLGN. **h** Epifluorescence images of CSPG4 immunofluorescent labelling in the adult intact LGN (co-labelled with CTB). **h'** Confocal images of CSPG4 and different cell populations markers (GFAP, Iba1, Olig2 and NeuN) in the adult intact dLGN and vLGN. **d, e, f, g, h** scale bar: 200 μm. **d', e', f', g', h'**: scale bar: 50 μm. All images are representative of N = 3 biologically independent animals.

does not modify the guidance landscape of the distal choice point (the optic chiasm) and brain targets. This suggests that guidance factors are stably expressed in the mature brain and may contribute to the guidance defects observed during axon regeneration.

We then wondered whether establishment of this guidance map is concomitant with the initial innervation of the target regions during development. We focused on the dLGN as a proof-of-concept, as its development and innervation and maturation have been thoroughly studied[49]. We looked at different time points of its innervation by RGC axons. To track dLGN innervation during development, we injected CTB555 in one eye of mice from E15.5 to P14 (Fig. 5a). Contralateral RGC axons enter the dLGN between E16 and P0, while ipsilateral RGC axons enter between P0 and P2, with synapse refinement and eye-specific segregation of innervation territories happening at eye opening (around P12)[50].

We looked at the dynamics of expression of different families of guidance molecules in the dLGN throughout development, that were detected in the adult dLGN in our analysis (Fig. 5b–f and Supplementary Fig. 4a, b). Using immunofluorescence and Western blot quantitative analysis, we found that each guidance cue displays a dynamic regulation of expression over the time of circuit formation. In particular for CSPG4 and Ephrin-B3, known to have a repulsive guidance activity, we observed that their level of expression is low at early stage of development (up to P2 for CSPG4 and up to P10 Ephrin-B3) (Fig. 5b, f). For CSPG4, expression gradually increases at times of synaptogenesis and circuit refinement (around P6) and stabilizes at times of visually-evoked activity (around P14) (Fig. 5b, Supplementary Fig. 5a, b). For Ephrin-B3, expression was undetected before P10, then it increases at times of circuit consolidation and visually-evoked activity (around P14) (Fig. 5f, Supplementary Fig. 5a, b). The cell adhesion molecule NCAM1 known to induce neurite growth[23] shows gradual increase at the time of dLGN innervation in the mouse embryo (E16 to P2). Then NCAM1 shows a progressive decrease until stabilization from P6 and throughout adulthood (Fig. 5c, Supplementary Fig. 5a, b). The gradual increase of Sema7A in the developing dLGN (Fig. 5e, Supplementary Fig. 5a, b) is also consistent with what was observed during neuronal circuit formation in multiple CNS regions[51]. In adult, Sema7A stabilized expression may play a role in neuroglia interactions and plasticity[52].

In contrast, we found that the adult guidance maps in the dLGN are not modified after optic nerve injury. For the candidate molecules analyzed above, we quantified their expression level by Western blot and found no significant difference in intact versus injured conditions (Supplementary Fig. 5c-d), which confirms our MS-based proteomic results. In the context of adult regeneration, our analysis of developmental stages supports the idea that guidance factors are dynamically regulated during circuit formation – particularly during innervation of the functional targets. In adult, the fact that guidance factors are unchanged upon optic nerve injury correlates with the failure of reinnervation of visual targets by regenerating axons. Our analysis provides a comprehensive map of guidance cues expression in the mature visual system (Fig. 6). This brings up the possibility that a tight

spatio-temporal regulation of axon guidance processes in adult should control regenerative axons navigation to their proper targets to resume neuronal functions.

### Mature regenerative RGC axons have the machinery to integrate guidance signaling

Based on our description of the guidance landscape in the mature brain, we then asked whether RGC had the potential to integrate and to respond to these signals. Therefore, we analyzed the expression of corresponding guidance receptors in RGC. To this end, we used available transcriptomic datasets of two recent single-cell studies[53,54] and of two regenerative models: co-activation of mTOR and JAK/STAT pathways[55] and overexpression of Sox11[56]. Using our map of guidance ligands expressed in mature visual targets (Fig. 2a), we explored expression of the corresponding guidance receptors. We found that corresponding receptors are expressed in RGC both in intact WT condition and in regenerative (post-injury) condition. For example, the transcripts of Ephrin-B3 receptors *EphB2* and *EphA4* are expressed in RGC in intact and in regenerative (post-crush) conditions (Supplementary Fig. 6a). In the intact datasets, we highlighted the number of RGC clusters showing expression of the receptors of interest. For example, the transcript for the Sema4D receptor *Plxnb1* is detected in 27 out of 45 subpopulations of adult RGC (Supplementary Fig. 6a). In fact, despite compelling evidence of differential injury response of neuronal subpopulations[54,57], almost all RGC clusters express guidance receptors for CSPG family members, Ephrin-B3, NCAM1 and 2, Sema4D and Sema7A, suggesting that adult guidance response is shared by all RGC subpopulations.

Furthermore, we looked at the variation of axon guidance molecules between regenerative conditions and non-regenerative (WT) conditions following injury[55,56]. Interestingly, some guidance receptors are dynamically regulated (either up or down) in the regenerative conditions, eg *Robo1* and *Plxna1* strongly upregulated compared to WT (Supplementary Fig. 6b). This suggests that regenerative RGC may modify their guidance response in these models. We also observed that some guidance receptors, such as *Epha7* and *Epha3*, vary in the opposite directions in the two regeneration models, probably due to the differences in pathway activation. Altogether, these observations account for the complexity of RGC guidance response that has to be considered to correct their misguidance.

### Mature regenerative RGC axons respond to guidance cues

Next, we used the guidance map we generated (Fig. 6) to determine the guidance modalities in the mature brain. We focused on the optic chiasm, as a major intermediate target during circuit formation[58]. We sought to determine the potential guidance role of two pairs of ligand/receptor: Ephrin-B3/EphA4[59,60]-EphB2[61,62], and Sema4D/Plexin-B1 during midline crossing. Ephrin-B3 is repulsive to commissural axons during midline crossing of the developing spinal cord[62] and acts as a growth-inhibitory barrier for regenerating axons at the lesion site[60]. Sema4D has been reported to induce cell morphological changes notably growth cone collapse by interacting with Plexin-B1[63,64] and

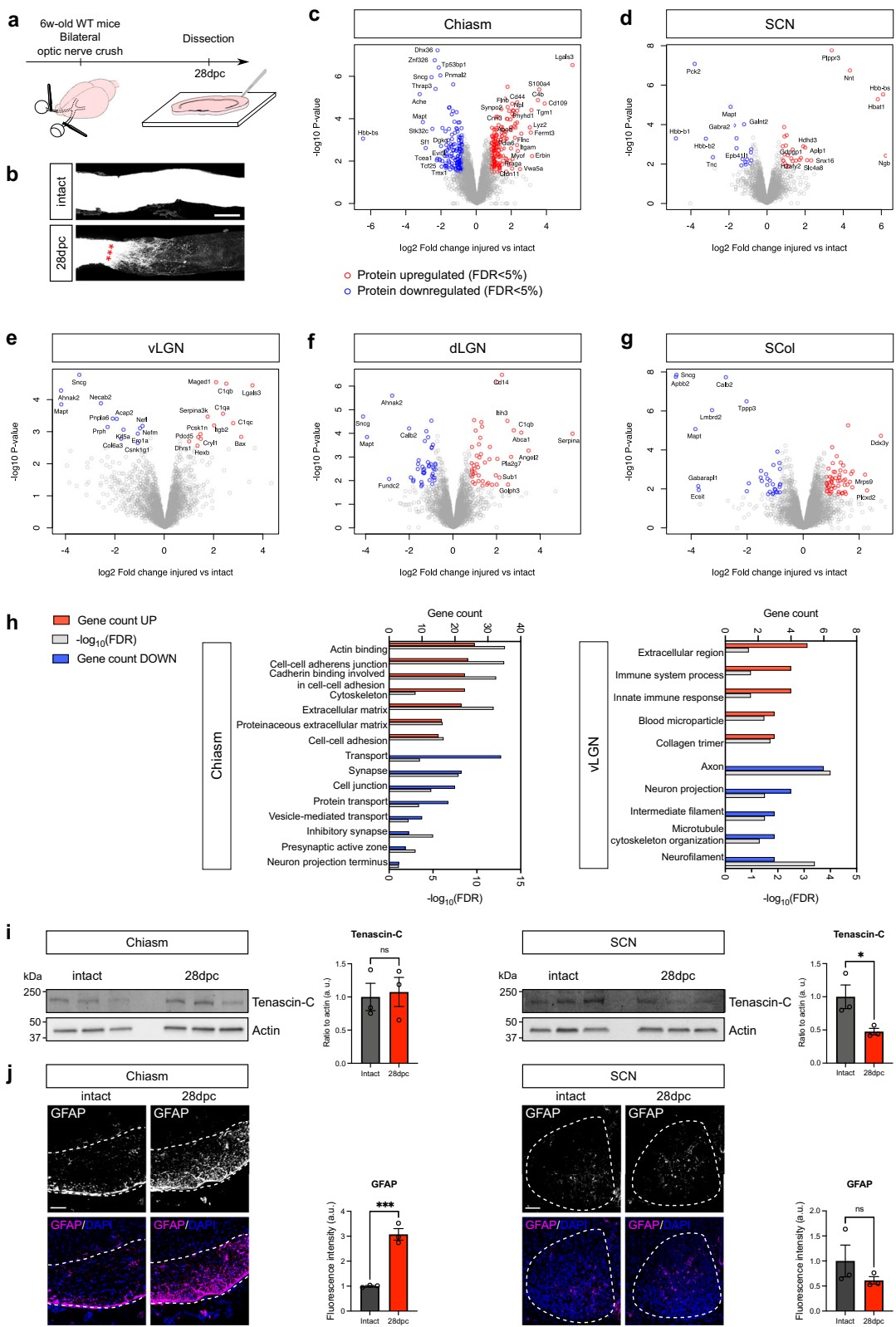

contributes to CNS regeneration failure at the lesion site[65]. These features make these two candidates relevant to challenge to study axon guidance in the optic chiasm.

First, we verified their expression by immunostaining on chiasm sections (Fig. 7a). We found that Ephrin-B3 and Sema4D are expressed in the chiasm, which makes these factors relevant for the question of chiasm entering and midline crossing. We used in situ hybridization to confirm *EphA4, EphB2* and *Plexin-B1* expression in the RGC layer of retina sections of wild-type mice, and in PTEN/SOCS3-co-deleted RGC at 3 days and 28 days after injury (Fig. 7b). Additionally, we used immunofluorescence to label the expression of these receptors in individual growth cones of RGC axons from adult retina explant cultures (Fig. 7c). Our results suggest that mature axons have the tools to interact with guidance cues and integrate their signal.

**Fig. 3 | Optic nerve crush causes modifications in the proteome of adult visual targets. a** Experimental design and timeline. 6 week-old mice underwent bilateral optic nerve crush and brains were dissected 28 days after the injury (28 dpc). **b** Cleared optic nerve labelled with CTB-Alexa555 in intact and injured conditions, allows to verify the efficiency of optic nerve crush. Red stars indicate the injury site. Scale bar: 100 μm. Data are representative of $N$ = 4 biologically independent animals. **c–g** Volcano plots showing differentially expressed proteins in the optic chiasm, SCN, vLGN, dLGN and SCol in injured versus intact conditions. Samples from visual targets at 28 days post-crush (28 dpc) were compared with the intact condition by MS-based label-free quantitative proteomics, using four biological replicates for each condition. The injury leads to modifications of the proteome of the visual targets, with 311 proteins found differentially expressed in the optic chiasm, 45 in the SCN, 26 in the vLGN, 92 in the dLGN and 92 in the SCol. Statistical testing was conducted using limma test. Differentially-expressed proteins were sorted out using a $\log_2$(fold change) cut-off of 0.8 and a $p$-value cut-off allowing to reach a FDR inferior to 5% according to the Benjamini-Hochberg procedure. **h** GO terms (DAVID analysis) associated with lesion-modulated hits in the optic chiasm

and in the vLGN. In red is represented the protein count of the corresponding GO for up-regulated proteins. In blue is represented the protein count of the corresponding GO for down-regulated proteins. In grey is represented the significance as $-\log_{10}$(FDR). **i** Western blot analysis of Tenascin-C expression on independent biological replicates for intact and 28dpc optic chiasm and SCN (left) and corresponding quantification (right). Each lane corresponds to tissue collected from one animal. For each sample, protein expression is quantified by pixel densitometry relative to actin and normalized to intact condition. $N$ = 3 biologically independent animals. Data are presented as mean values +/−SEM. Two-tailed unpaired Student's $t$-tests, ns: not significant, *$p$-value = 0.0464, ns: not significant. **j** Epifluorescence images of immunofluorescent labelling on coronal brain section showing the modulation of GFAP expression in the optic chiasm and not in the SCN in intact and injured conditions (left) with corresponding quantification (right). Scale bar: 50 μm. $N$ = 3 biologically independent animals. Data are presented as mean values +/− SEM. Two-tailed unpaired Student's $t$-tests, ns: not significant, ***$p$-value = 0.0009, ns: not significant. Source data are provided as a Source Data file.

To address axon responsiveness to these guidance cues, we combined the ex vivo system of culture of adult retina explants[66] with a stripe assay[67] (Fig. 7d). This type of assay has been originally used to characterize the guidance properties of developing neurons to various extracellular matrix proteins, short-range cues and cell adhesion molecules. Silicon matrices are used to create a striped pattern of substrates on which explants are plated. In our experiment, we assessed the preferential outgrowth of adult RGC axons on a guidance cue fused to Fc fragment versus Fc only (Fig. 7e). The preference was quantified as the number of outgrowth points in the alternating stripes.

In the control condition, many axons grow out of the explant with no preference on the stripes. In contrast, for Ephrin-B3- and Sema4D-coated stripes, we found that adult RGC axons tend to avoid Ephrin-B3-positive stripes and Sema4D-positive stripes, while outgrowing preferentially on Fc-stripes alone (Fig. 7e). Moreover, there are fewer and shorter axons in these conditions, further highlighting the repulsive effect of these cues. As a control of specificity, we found that the guidance cue Sema7A is neutral to adult RGC outgrowth (Fig. 7e), despite expression of its receptor Plexin-C1[51,68] in growth cones ex vivo and in RGC in vivo (Fig. 7b, c). Together, these ex vivo data show that adult RGC axons are able to integrate Ephrin-B3 and Sema4D signaling and to respond to these cues.

### Ephrin-B3 and Sema4D control midline crossing of regenerating axons in the mature chiasm

Ephrin-B3 and Sema4D are expressed in the chiasm and adult RGC axons respond to these cues. We then sought to determine whether modulation of the guidance signaling could impact midline crossing of regenerating axons in vivo. To this end, we used a silencing approach of the guidance receptors, by targeting the receptors of Ephrin-B3 and Sema4D with shRNA – EphA4/EphB2, and Plexin-B1, respectively. For each shRNA, we verified the silencing in vivo using in situ hybridization on adult retina sections (Supplementary Fig. 7a).

We first analyzed the outcome of inhibition of these receptors in wild-type condition. To do so, we injected intravitreally an AAV expressing shRNA scrambled (sh scrambled), sh EphA4, sh EphB2 or sh Plexin-B1 (Supplementary Fig. 7b). In control condition (AAV-sh scrambled), no regeneration is observed 14 days after optic nerve injury (Supplementary Fig. 7c). Interestingly, we found that both the combination of sh EphB2 and sh EphA4, and the sh Plexin-B1 had a significant effect on regeneration, as quantified by the number of regenerative axons (Supplementary Fig. 7c). Regarding Ephrin-B3 signaling, this result is consistent with the regenerative phenotype observed in Ephrin-B3-null mice both in the optic nerve and in the spinal cord[69], as well as with the enhanced regenerative effect in EphA4-null mice in the optic nerve[70]. This experiment shows that it is

necessary to overcome the barrier to axon growth at the lesion site itself, a phenomenon widely studied in various CNS injury models.

Next, we assessed axon guidance at the optic chiasm by combining the guidance receptor silencing approach with a long-distance regeneration model through the co-activation of mTOR and JAK/STAT pathways in RGC[55]. In this model, regenerative axons reach the optic chiasm at 28 days post-crush (28dpc). Here, we injected the AAV2 expressing the shRNA against the receptor (or sh scrambled as a control) in one eye of 3 week-old Pten[fl/fl] SOCS3[fl/fl] mice (Fig. 8a). Then one week later, we activated the pro-regenerative pathways by injecting AAV2-Cre recombinase and AAV2-CNTF in the same eye. We performed optic nerve injury two weeks later. We then assessed the regeneration extent and the effect on midline crossing at 28 days post-crush by injecting cholera toxin B in the regenerative eye. Our results show that, while the number of regenerative axons (reaching the distal part of the regenerating optic nerve) is unchanged by receptor silencing, their trajectory when entering the chiasm is modified. Both EphA4/EphB2 and Plexin-B1 silencing led to a higher number of axons crossing the midline, and a higher number of axons into the optic chiasm, compared to sh scrambled condition (Fig. 8b).

Altogether, our results show that i) adult regenerative RGC axons are able to respond to Ephrin-B3 and Sema4D signaling in vivo; and ii) these cues alter chiasm entering and midline crossing.

## Discussion

Recent advances in the field of CNS regeneration have led to long-distance regeneration[3,4,44,55,71]. Yet, this achievement comes with an unexpected drawback as most regenerative axons are misguided away from their proper targets, counteracting any attempt of functional recovery[3,4,55]. This brings out axon guidance as a key process in the adult CNS to rebuild a functional circuit following injury. As of today, there are few data regarding axon guidance in the mature brain, and many questions regarding the potential and relevance of guiding axons following injury remain. In the present study, we performed extensive proteomic characterization of the optic chiasm and of the key brain targets receiving RGC inputs. We demonstrated that axon guidance modalities are still active in adult. We generated a comprehensive map of guidance cues expression in the visual system (Fig. 6) that explains guidance defects widely observed during regeneration. As a proof-of-concept, we manipulated two guidance signalings identified from our screen: Ephrin-B3 and Sema4D. Thereby, we showed that these cues are essential for regenerating axons to enter the chiasm and to cross the midline in the optic chiasm. Altogether, our results highlight that mature axons can be guided to control their navigation. In the context of regeneration, this feature is essential to form a functional circuit.

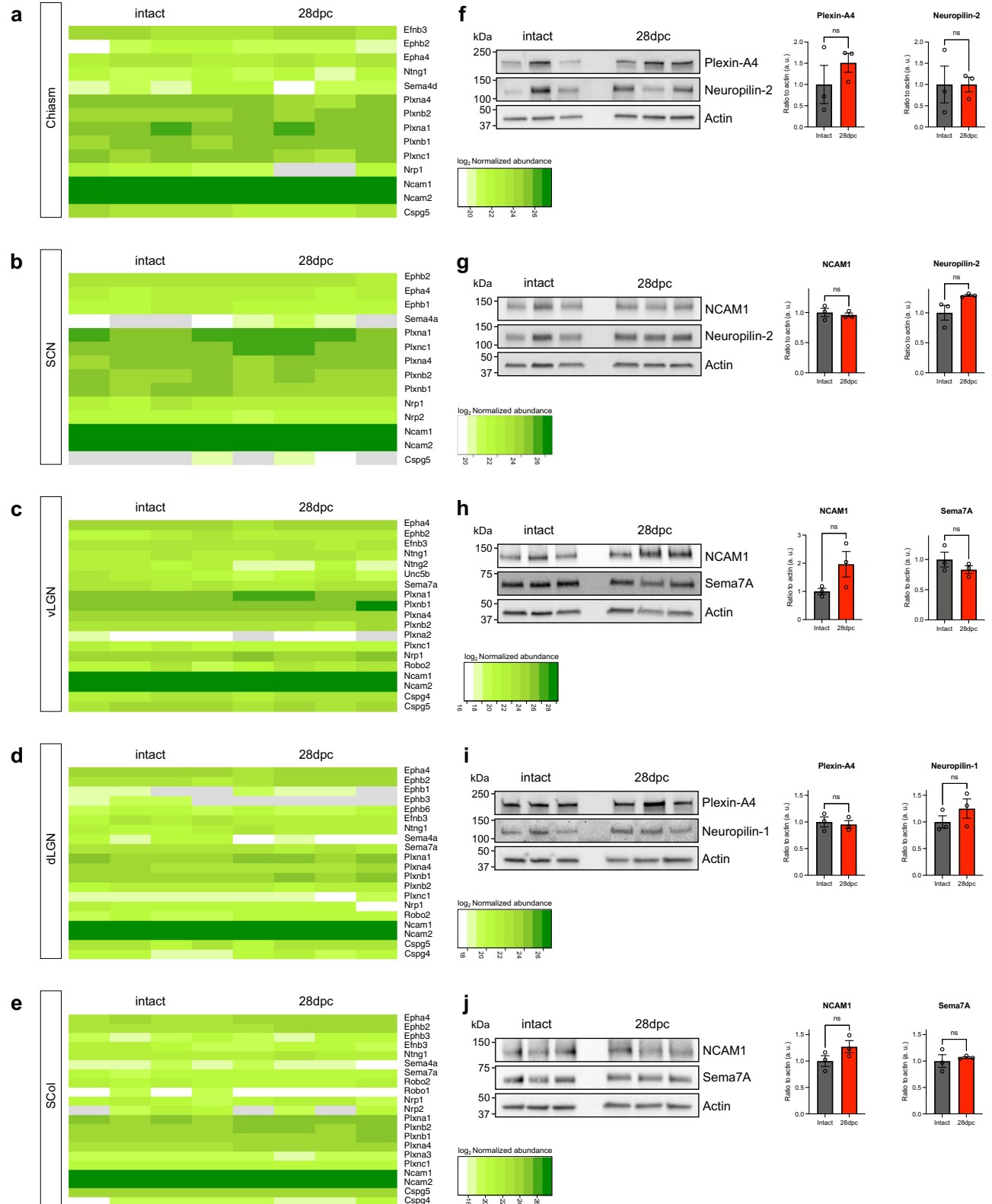

**Fig. 4 | Expression of guidance and guidance-associated factors remains steady in the adult brain visual targets after optic nerve injury. a–e** Heatmaps showing protein expression levels (log₂ Normalized abundance) in all replicates of intact and injured (crush) conditions. **f–j** Western blot analysis of selected proteins on independent biological replicates of each visual targets (left) and corresponding quantification (right). Each lane corresponds to tissue collected from one animal. For each sample, protein expression is quantified by pixel densitometry relative to actin and normalized to intact condition. $N = 3$ biologically independent animals. Data are presented as mean values +/− SEM. Two-tailed unpaired Student's $t$-tests, ns: not significant. Source data are provided as a Source Data file.

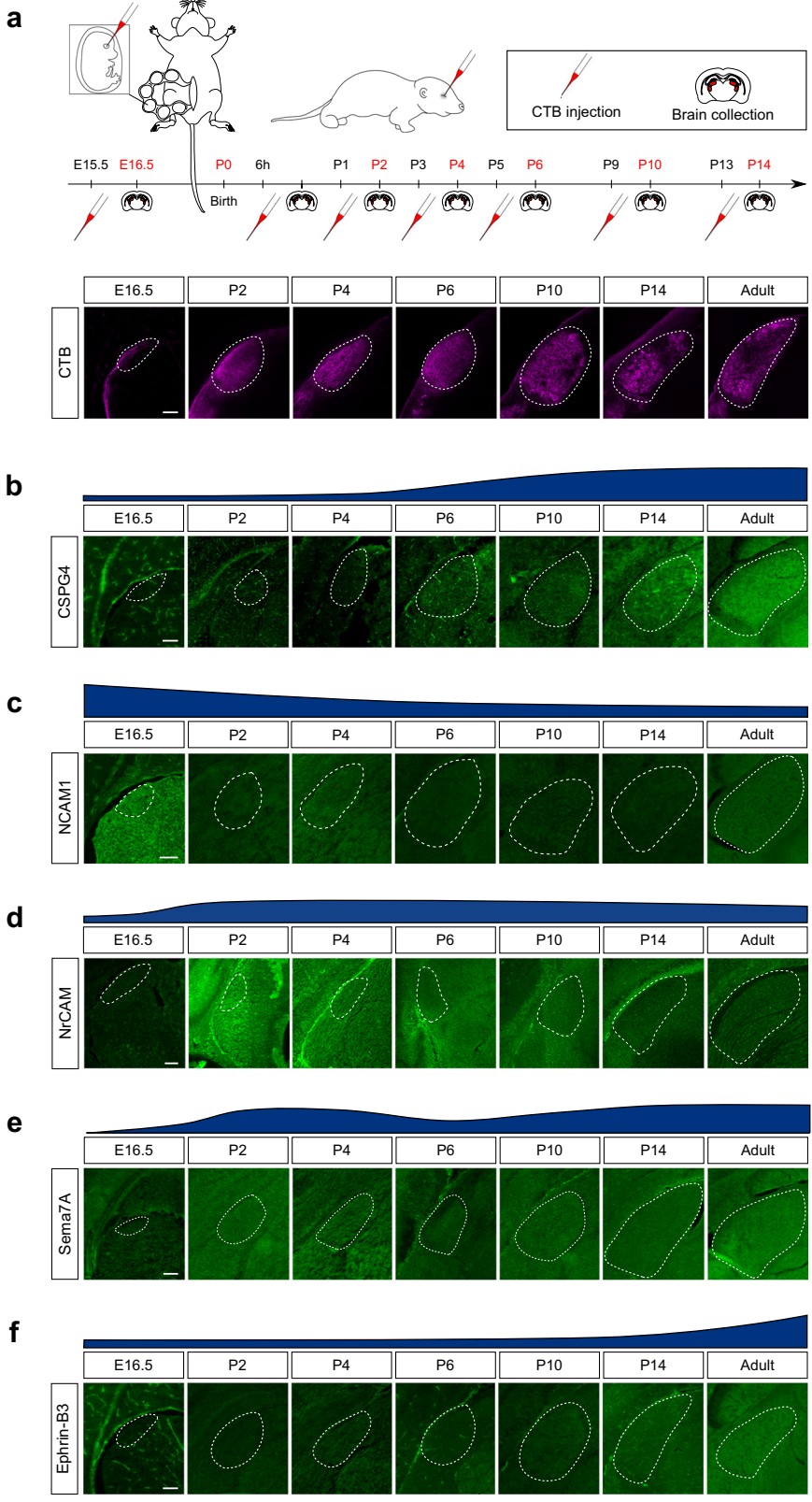

**Fig. 5 | The guidance map is established during innervation of the developing visual target regions. a** Schematic drawing showing in utero injection of mouse embryo eye at E15.5 and postnatal eye injection. Experimental design and timeline, and representative epifluorescent images of CTB-labelled dLGN at the different time points. For P2 to P14 time-points: one day before sample collection, one eye of mouse pups was injected intravitreally with CTB-555. For P0 time-point: 6 h before sample collection, one eye of mouse pups was injected intravitreally with CTB-555. **b**–**f** Epifluorescent images of immunofluorescent labelling of **b** CSPG4, **c** NCAM1, **d** NrCAM, **e** Sema7A and, **f** Ephrin-B3 in the developing dLGN (co-labelled with CTB, marked with white dashed lines on coronal sections). Scale bar: 100 μm. All images are representative of *N* = 3 biologically independent animals.

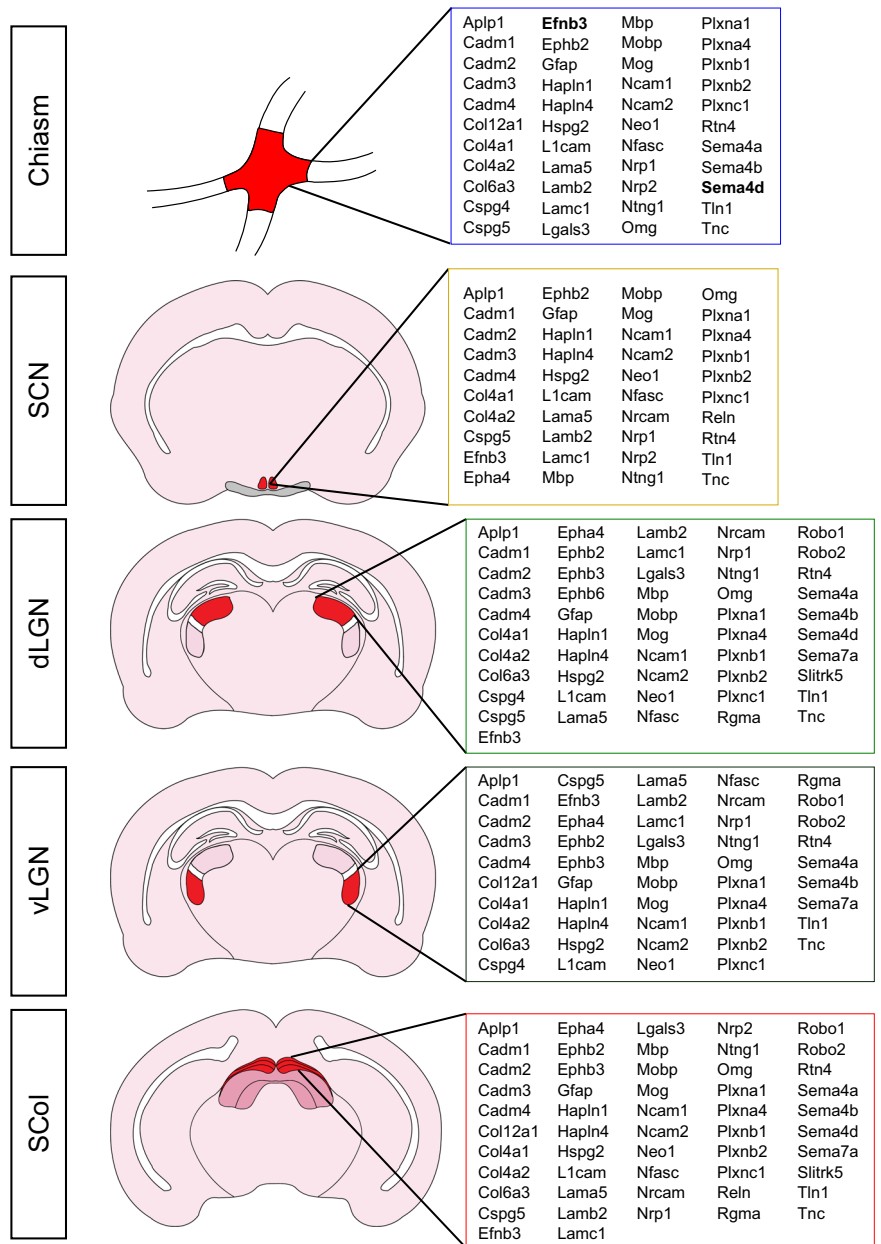

**Fig. 6 | Guidance maps in the adult brain targets are set during development and provide insight into the guidance defects encountered by mature regenerative axons.** Summary of guidance maps as described in the present proteomic study for each brain target of the adult visual system. Proteins of interest are in bold. SCN suprachiasmatic nucleus, LGN lateral geniculate nucleus, v ventral, d dorsal, SCol superior colliculus.

Axon guidance is a process extensively studied during development[72], particularly at intermediate targets, such as the floorplate in the spinal cord[73] or the optic chiasm in the visual system[74]. In the mature nervous system, guidance process may not be required anymore, as the circuits have reached a steady state where axon growth is very limited. Some guidance cues have been involved in other processes, such as plasticity and regulation of synaptic activity[35–37]. However, the comprehensive map of their expression and regulation in the adult brain is not known, and neither is their potential for guidance activity on adult regenerating axons. In this study, we conducted a proteomic analysis on the mature visual system, with a particular focus on expression of guidance cues and guidance-associated factors involved in axon pathfinding. We found that several proteins are expressed by all targets, reflective of the shared guidance signature of RGC targets (eg CSPG5, NCAM1 and 2, members of the Plexin family). Most guidance cues are associated with neurons, as

we showed for NCAM1 or CSPG4. Other studies addressing the changes within the LGN during development have shown that these molecules are also associated with neurons and more precisely with perineuronal nets (PNN)[31]. In our case, the pattern of expression suggests that guidance cues may be associated with PNN. Interestingly, we found that Sema4D expression is associated with oligodendrocytes, as previously reported[65,75]. Moreover, depending on the brain regions, some cues are associated with one or another cell type. For example, NCAM1 is associated with GFAP+ astrocytes in the chiasma whereas it is expressed by neurons in the SCN, vLGN, dLGN and SCol. These results suggest that these cues may have different functions depending on the brain region and their source. Indeed, cell adhesion molecules have been shown to regulate axon fasciculation when expressed by neurons[76]. Beside their role in adhesion, CAMs are also involved in modulating axonal response to guidance cues[77,78]. Interestingly, during development, growth cones from pioneer axons interact with glia cells

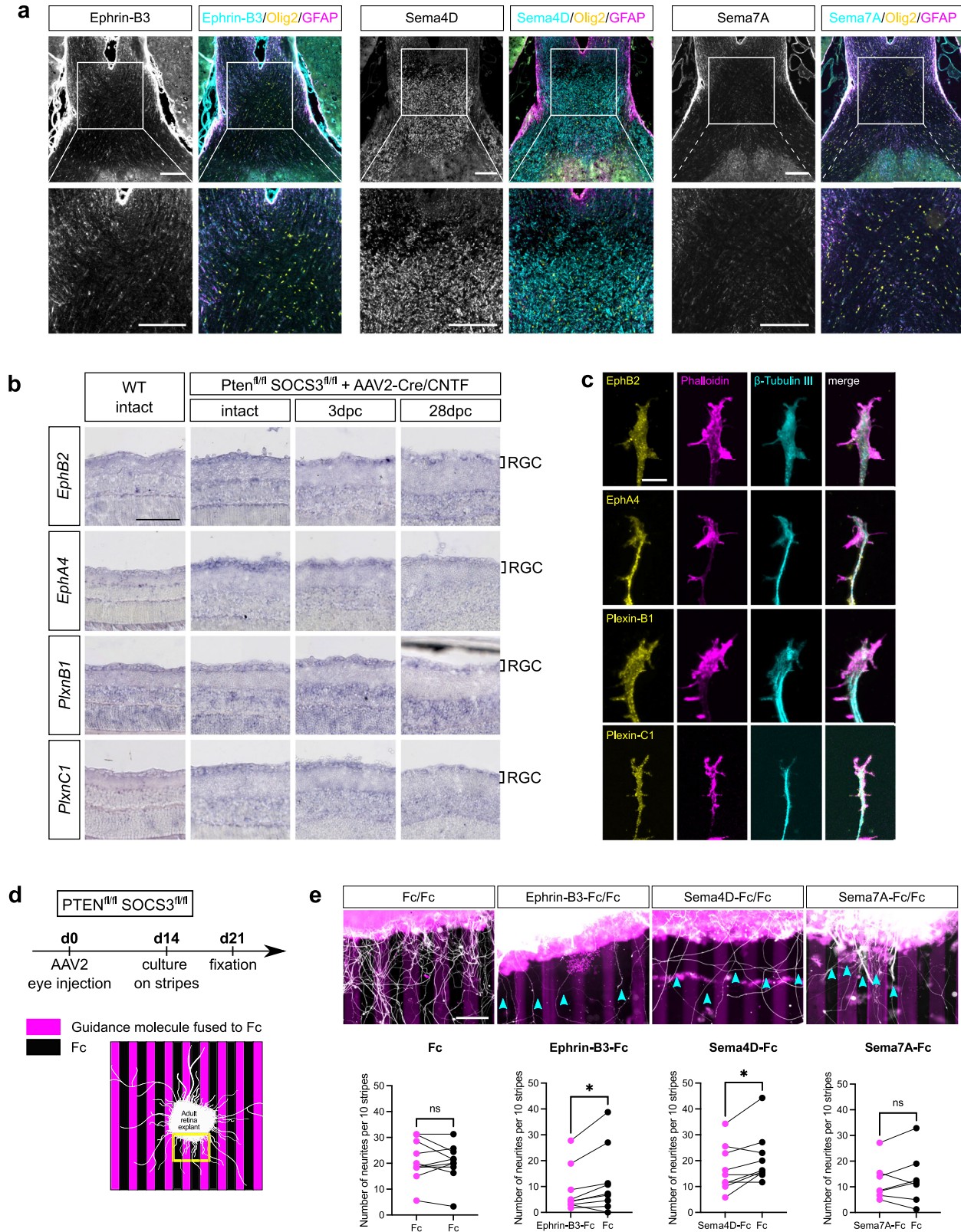

at critical choice points[79] and CAM, specifically NCAM stands as one of the major effectors during this navigation[80]. Thus, the expression of NCAM1 by astrocytes at the optic chiasm in the mature brain may be important for axon navigation during regeneration at the midline.

In the context of CNS regeneration, guidance cues have been mainly studied at the lesion site, where their upregulation upon injury correlates with axon growth inhibition[26,65,69,81]. In contrast, their

regulation distal to the lesion site or along the axonal path has not been addressed. Thus, we compared their expression in intact condition versus bilateral optic nerve injury at 28dpc. Surprisingly, while we observed several protein expression changes in the targets of interest, the expression of guidance cues themselves was not affected by the removal of RGC afferences. These results suggest that injury signals are integrated by the brain nuclei and the optic chiasm, and that guidance

**Fig. 7 | Axons respond to guidance factors ex vivo. a** Epifluorescence images of immunofluorescent labelling of Ephrin-B3, Sema4D and Sema7A in transversal optic chiasm sections, with counter-staining of Olig2 and GFAP. Scale bar: 200 μm. Images are representative of $N = 3$ biologically independent animals. **b** In situ hybridization showing EphB2, EphA4, Plexin-B1 and Plexin-C1 expression in RGC of WT intact, PTEN^fl/fl/SOCS3^fl/fl intact, 3 days post-crush (3dpc) and 28 days post-crush (28dpc) retina infected with AAV2-Cre and AAV2-CNTF. Scale bar: 100 μm. Images are representative of $N = 4$ eyes. **c** Representative confocal immunofluorescence pictures showing expression of EphB2, EphA4, Plexin-B1 and Plexin-C1 in the growth cone of a PTEN^-/-/SOCS3^-/- RGC axon. Scalebar: 5 μm. Images are representative of $N = 3$ explants over one experiment. **d** Timeline and experimental design of the stripe assay. Yellow scare represents an example of region of interest used for the quantification. 14 days after AAV2-Cre and AAV2-CNTF eye injection in PTEN^fl/fl/SOCS3^fl/fl mice, retina explants were cultured for 7 days on the stripes pattern containing a guidance molecule fused to Fc or Fc as a control. **e** Epifluorescence images of retina explants cultured on stripes containing Fc/Fc (as a control), Ephrin-B3-Fc/Fc, Sema4D-Fc/Fc and Sema7A-Fc/Fc with the corresponding quantification. Blue arrowheads highlight axons path. Fc/Fc: $N = 10$ explants over two independent experiments. Ephrin-B3-Fc/Fc: $N = 9$ explants over two independent experiments. Sema4D-Fc/Fc: $N = 9$ explants over two independent experiments. Sema7A-Fc/Fc: $N = 7$ explants over two independent experiments. Two-tailed paired t-tests, Ephrin-B3-Fc/Fc *p-value = 0.0343, Sema4D-Fc/Fc *p-value = 0.024, ns: not significant. Scale bar: 100 μm. Source data are provided as a Source Data file.

cues expression is not affected by these changes. The analysis of the guidance map establishment during development of the dLGN supports the idea that guidance factors are dynamically regulated during circuit formation – particularly during innervation of the functional targets. In adult, the fact that guidance factors are unchanged upon optic nerve injury correlates with the failure of reinnervation of visual targets by regenerating axons. So, our analysis provides additional argument that, following injury, the adult CNS does not have the appropriate guidance map that is required for functional reconnection. This brings up the possibility that a tight spatio-temporal regulation of axon guidance processes in adult should allow axons to navigate correctly and enter the proper target to resume neuronal functions.

In our study, we also analyzed the injury response of the primary targets of the lesioned RGC axons. Previous transcriptomic work focused on gene expression changes in the superior colliculus in the course of development and following injury, based on mononuclear enucleation[82]. This study highlighted functional groups of genes regulated by the injury, including transport and metabolism. More recently, a transcriptomic profiling of the spinal cord target tissue after stroke allowed to highlight two phases of the target injury response: a primary phase characterized by the inflammatory response in the target area, and a secondary later phase characterized by secretion of connectivity-promoting growth factors[83]. Such changes in the long-term might affect axon growth and guidance of regenerative axons and open a therapeutic window of modulation of gene expression to promote growth and synaptogenesis. Intriguingly, we uncovered protein expression changes after injury, even in distal targets. Multiple hypotheses arise here: i) target neurons that are not sustained anymore by RGC neuronal activity undergo protein expression changes; ii) molecular changes of the distal response may be intrinsic to these target neurons and possibly consequent to forward signalling of axonal injury; iii) axon degeneration down to the lesion site triggers a local inflammatory response which in turns causes injury response of the neuronal and non-neuronal cell populations with adaptation of protein expression. It is more expected that the latter is a transient response short to mid-term following the injury, as observed in the Wallerian degeneration model[84]. In our case, four weeks after the injury, we expect that axon death and debris have been cleared by glial cells. Yet, sustained activation of the glia and activation of the immune system are a feature of CNS injury[85], consistent with the enrichment of immune response in the pool of upregulated proteins in several targets. Even if these events might not be involved in axon guidance, they should be considered when it comes to circuit formation, as they could interfere with synapse formation, maintenance or myelination of regenerative axons.

As guidance cues are still expressed all along the path of regenerating axons, we asked whether axon guidance is still functional in the mature brain. Our ex vivo experiments stress out that regenerating axons can respond to guidance signaling, that this response is specific and that it may be regulated over time. For example, one can hypothesize that regenerating axons do not respond to Sema7A at early steps of their growth, but eventually acquire sensitivity to it. This type of regulation has been thoroughly described during development of the spinal cord, where growing commissural axons do not respond to Sema3B or Sema3F before crossing the floorplate, despite expressing the receptor Neuropilin-2[86]. Only after reaching the midline do they become sensitive to these cues, as the Neuropilin-2 co-receptor Plexin-A1 is addressed to the membrane[78]. Such mechanism may be at play in axon regeneration in the mature brain.

We then addressed in vivo axon growth and guidance during regeneration in adult. To this end, we focused on the optic chiasm as this region is a critical choice point[24], where many guidance defects are observed in long-distance regeneration models. In the intact visual circuit, 95% RGC axons cross the midline and 5% project ipsilaterally[24]. During regeneration, many axons do not cross the midline and get lost in the optic chiasm[10,55]. Here, we show that Ephrin-B3 and Sema4D are expressed in the optic chiasm and their corresponding receptors are expressed by RGC. These guidance cues have been described during the formation of the nervous system[87–89]. In the context of the mature nervous system, they are associated with a decrease of axonal outgrowth and are considered as inhibitory for axon regeneration[60,65,70,90]. Yet, their role as guidance cues of regenerating axons have not been addressed. Here, we combined a shRNA-based receptor-silencing approach with a long-distance regeneration model (mTOR/JAK-STAT activation[55]). In this case, we observed a significant modification of regenerating axons pathfinding during midline crossing at the optic chiasm. While the ipsi- versus contralateral segregation is precisely described during development, this specification remains to be determined in regenerating adult axons. In our case, it is possible that the repulsive effect observed using our adult assay affects the majority of RGC axons, with no segregation of ipsi- versus contralateral projections. Further analyses will be critical to elucidate the behavior of axons of each RGC subpopulation, the extent of their ability to resume developmental pathfinding, and whether these rerouted axons are able to form stable synapses with their original targets to sustain functional recovery. Nonetheless, we here demonstrate the possibility to guide regenerating axons in vivo to repair a functional circuit.

Altogether, our datasets provide an unbiased extensive protein screen of primary visual targets of the adult CNS, in intact and injured conditions. While axon guidance has been quasi-exclusively studied during development so far, we demonstrate here that it is in fact at play in the adult system. In particular, our study shows that: i) guidance cues are expressed in the adult central nervous system; ii) their expression is not modified upon injury; and iii) adult regenerating axons are able to respond to these guidance cues. In sum, we provide a proof-of-concept that guidance in the adult is possible and particularly relevant in the context of regeneration. This essential process will set the building blocks of circuit repair in the injured adult visual system.

## Methods
### Mice
Wild-type (C57BL/6) embryos (E16.5), pups (P0, P2, P4, P6, P10, P14) and adult (6 to 10 week-old) mice were used in this study, regardless of

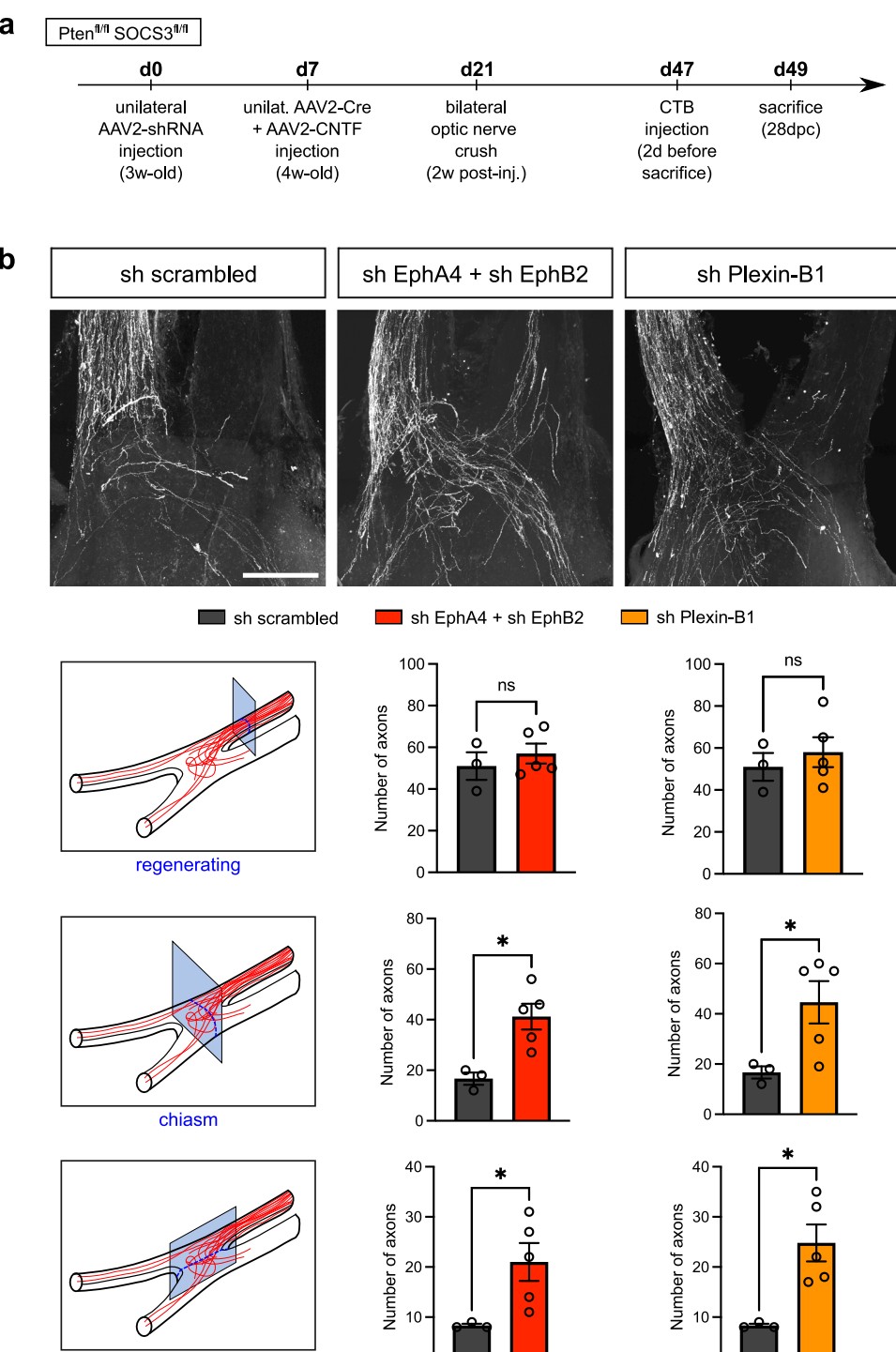

**Fig. 8 | Receptor silencing induces a modification in pathfinding of regenerating axons in PTENfl/flSOCS3fl/fl mice. a** Timeline of the experiment. 3 week-old PTEN$^{fl/fl}$SOCS3$^{fl/fl}$ mice were first injected with shRNA EphA4+ shRNA EphB2 or shRNA-Plexin-B1 or shRNA scrambled. 7 days after, AAV2-Cre and AAV2-CNTF eye injections were performed. 2 weeks post-injection, mice undergone bilateral optic nerve crush and phenotypes were observed 28 days post-crush (28dpc). **b** Whole optic chiasm confocal images showing regenerative axons labelled with CTB in PTEN$^{fl/fl}$SOCS3$^{fl/fl}$ mice injected with AAV2-shRNA, and AAV2-Cre + AAV2-CNTF at 28dpc, with schematic of quantification planes and corresponding quantification. Sh scrambled: $N = 3$ biologically independent animals. Sh EphA4 + sh EphB2: $N = 5$ biologically independent animals. Sh Plexin-B1: $N = 5$ biologically independent animals. Data are presented as mean values + /− SEM. Entering the chiasm: two-tailed unpaired Student's $t$-tests, shEphA4+sh EphB2 vs sh scrambled *$p$-value = 0.0129, sh Plexin-B1 vs sh scrambled *$p$-value = 0.049. Crossing the midline: two-tailed unpaired Student's $t$-tests, shEphA4+sh EphB2 vs sh scrambled *$p$-value = 0.0459, sh Plexin-B1 vs sh scrambled *$p$-value = 0.0154. Scale bar: 250 μm. Source data are provided as a Source Data file.

sex. PTEN$^{fl/fl}$/SOCS3$^{fl/fl}$ (C57BL/6 background) mice were used for co-culture and in vivo experiments. All the in vivo experiments were performed in accordance with our ethics protocol approved by the institution, local ethics committee and the French and European guidelines (APAFIS#9145-201612161701775v3 and APAFIS#26565-2020061613307385v3). The number of mice used in each experiment is specified in the corresponding figure legends.

## Ethics statement
All animal care and procedures have been approved by the Ethics Committee of Grenoble Institut Neurosciences (project number 201612161701775) and by the French Ministry of Research (Ministère de l'Enseignement Supérieur, de la Recherche et de l'Innovation, project numbers APAFIS#9145-201612161701775v3 and APAFIS#26565-2020061613307385v3) in accordance with French and European guidelines.

## Optic nerve crush injury
Optic nerve crush was performed according to ref. 66. Briefly, 6 week-old mice were anesthetized with intraperitoneal injection of ketamine (60–100 mg/kg) and xylazine (5–10 mg/kg). A mini bulldog serrefines clamp was placed to display the conjunctiva. The conjunctiva was incised lateral to the cornea. The refractor bulbi muscles were gently separated and the optic nerve was pinched with forceps (Dumont #5 FST) at 1 mm from the eyeball during 5s[2,66].

## Intravitreal injection
**Intravitreal CTB injection.** One day before sample collection, mice were anesthetized as described above and intravitreal injections of CTB-555 (Cholera toxin subunit B, Alexa Fluor 555-conjugated, Thermo Fisher Scientific) were performed. The external edge of the eye was clamped using a mini bulldog serrefines clamp (FST) to display the conjunctiva. 1 μl of CTB-555 (1 mg/mL) was injected into the vitreous body using a glass micropipette connected to a Hamilton syringe[2,66].

**Intravitreal AAV2 injection.** Intravitreal Adeno-Associated Viruses of serotype 2 injections were performed as described above. For the PTEN SOCS regenerative model, AAV2 expressing the Cre recombinase (AAV2-Cre) and ciliary neurotrophic factor (AAV2-CNTF) were used. For the in vivo receptors' inhibition, the following AAV2 were injected: shRNA scrambled, shRNA Plexin-B1, shRNA EphB2, shRNA EphA4. The shRNA sequences are the following: shRNA scrambled GCTTAC-TAACCTCGGCAGT, shRNA Plexin-B1 GTATATCAACAAGTACTAT (Addgene, 115174)[91], shRNA EphB2 GGACCTTGTTTATAACATCAT[92,93], shRNA EphA4 GCAGCACCATCATCCATTG[94].

## Adult sample collection
Adult mice were subdivided into 2 groups: control (intact mice) and mice that underwent optic nerve injury. 10 week-old mice from both groups were anesthetized using isoflurane (Equipement Vétérinaire Minerve). After cervical dislocation, eyeballs and brains were dissected out. Using a vibratome (Leica VT1200S) 300 μm fresh brain sections were collected in ice-cold Hibernate A (Thermo Fisher Scientific). Chiasm, suprachiasmatic nucleus (SCN), ventral lateral geniculate nucleus (vLGN), dorsal lateral geniculate nucleus (dLGN) and superior colliculus (SCol) were micro-dissected under a binocular microscope (Zeiss SteREO Discovery.V8) and flash frozen in dry ice. Eyes of control mice were previously injected intravitreally with CTB-555 to facilitate detection of the visual targets.

## Sample preparation
Total protein lysates were obtained by extraction in Laemmli 2x (4% SDS, 60 mM TrisHcl pH 6.8, 200 mM DL-Dithiothreitol) and 15 min incubation on ice. 0.5 μl of benzonase (Sigma-Aldrich) was added to each sample and incubated for 10 min at 37 °C to digest DNA. After

denaturation at 95 °C for 5 min, the protein concentration was determined using Pierce BCA Protein Assay Kit (Thermo Fisher Scientific). To ensure sample quality and consistent quantification, a silver staining was performed by loading 1 μg of proteins onto a 4–15% precast gel (Biorad). After electrophoresis, silver staining of the proteins was performed using the Silver Quest Kit (Thermo Fisher Scientific).

## MS-based proteomic analyses
Proteins from tissue preparations were solubilized in Laemmli buffer before being stacked in the top of a 4–12% NuPAGE gel (Life Technologies), stained with R-250 Coomassie blue (Bio-Rad) and in-gel digested using modified trypsin (sequencing grade, Promega) as previously described[95]. The dried extracted peptides were resuspended in 5% acetonitrile and 0.1% trifluoroacetic acid and analyzed by online nanoliquid chromatography coupled to tandem mass spectrometry (LC–MS/MS) (Ultimate 3000 RSLCnano and the Q-Exactive HF, Thermo Fisher Scientific).

**Chiasm and SCN samples.** Peptides were sampled on a 300 μm 5 mm PepMap C18 precolumn (Thermo Fisher Scientific) and separated on a 75 μm 250 mm C18 column (Reprosil-Pur 120 C18-AQ, 1.9 μm, Dr. Maisch HPLC GmbH). The nano-LC method consisted of a 240 min multi-linear gradient at a flow rate of 300 nl/min, ranging from 5 to 33% acetonitrile in 0.1% formic acid.

**dLGN, vLGN and SCol samples.** Peptides were sampled on a 300 μm 5 mm PepMap C18 precolumn (Thermo Fisher Scientific) and separated on a 200 cm μPACTM column (PharmaFluidics, Ghent, Belgium). The nano-LC method consisted of a 360 min multi-linear gradient at a flow rate of 300 nl/min, ranging from 5 to 33% acetonitrile in 0.1% formic acid.

For all tissues, the spray voltage was set at 2 kV and the heated capillary was adjusted to 270 °C. Survey full-scan MS spectra (m/z = 400–1600) were acquired with a resolution of 60000 after the accumulation of $3 \times 10^6$ ions (maximum filling time 200 ms). The 20 most intense ions were fragmented by higher-energy collisional dissociation after the accumulation of $10^5$ ions (maximum filling time: 50 ms). MS and MS/MS data were acquired using the software Xcalibur 4.0 with configured instrument Q Exactive HF - Orbitrap MS 2.9 (Thermo Scientific).

## MS-based proteomic data processing
Data were processed automatically using Mascot Distiller software (version 2.7.1.0, Matrix Science). Peptides and proteins were identified using Mascot (version 2.6) through concomitant searches against Uniprot (Mus Musculus taxonomy, for Optic chiasm and SCN samples, downloaded in June 2019 with 87,573 entries, and for dlGN, vLGN and SCol samples, downloaded in August 2020 with 87,975 entries), a homemade database containing the sequences of classical contaminant proteins found in proteomic analyses (keratins, trypsin, bovine albumin, etc., 250 entries) and their corresponding reversed databases. Trypsin/P was chosen as the enzyme and two missed cleavages were allowed. Precursor and fragment mass error tolerances were set, respectively, to 10ppm and 25mmu. Peptide modifications allowed during the search were: carbamidomethylation (fixed), acetyl (protein N-terminal, variable) and oxidation (variable). The Proline software (version 2.0)[96] was used to merge either all intact brain targets together or intact and injured data together but for each brain target separately. After combination, results were filtered: conservation of rank 1 peptide-spectrum match (PSM) with a minimal length of 7 and a minimal score of 25. Peptide-spectrum matching (PSM) score filtering is then optimized to reach a False Discovery Rate (FDR) of PSM identification below 1% by employing the target decoy approach. A minimum of one specific peptide per identified protein group was set. For computing results of intact brain targets, Proline was used to

perform MS1-based label free quantification of the peptides and protein groups from the different samples without cross-assignment; protein abundances were computed as the sum of specific peptide abundances; for each tissue, an average iBAQ value[33] across replicates was then calculated. For producing quantitative results of intact brain targets *versus* injured ones, Proline was used to perform MS1-based label free quantification of the peptides and protein groups from the different samples with cross-assignment activated. After peptide abundances normalization, protein abundances were computed as a sum of specific peptide abundances.

## Statistical analysis of mass spectrometry-based proteomic data

Statistical analysis was performed using ProStaR[97] to determine differentially abundant proteins between intact and crushed conditions. Protein sets were filtered out if they were not identified in at least two replicates of one condition. Protein sets were then filtered out if they were not quantified across all replicates in at least one condition. Reverse protein sets and contaminants were also filtered out. After log2 transformation, POV missing values were imputed with slsa method and MEC ones with 2.5-percentile value of each sample. Statistical testing was conducted using limma test. Differentially-expressed proteins were sorted out using a $\log_2$(fold change) cut-off of 0.8 and a *p*-value cut-off allowing to reach a FDR inferior to 5% according to the Benjamini-Hochberg procedure. Only proteins identified with a minimum of two specific peptides were further considered.

## Data analysis

**Scatterplots.** Scatterplots of protein hits were obtained by plotting the protein abundances across replicates of a same visual target region in either intact or injured conditions.

**Gene Ontology analysis.** Gene Ontology (GO) analysis was performed using DAVID (Database for Annotation, Visualization and Integrated Discovery) Bioinformatics Resources (version 6.8). Bubble plots were obtained by submitting the list of the 3000 more abundant proteins (ranked by iBAQ, see Supplementary Data 1) of each brain target to DAVID. GO terms were divided into 3 groups: Biological Processes (BP), Molecular Functions (MF) and Cellular Compartments (CC). The following parameters were used as cut-offs to represent bubble plots: gene count >50, fold enrichment >3 and *p*-value < 0.01.

**Interactome analysis.** To obtain the protein-protein interaction networks, the 200 more abundant proteins (ranked by iBAQ, see Supplementary Data 1) were submitted to STRING (version 11.0). High confidence interactions (minimum required interaction score 0.700) were plotted with hiding of disconnected nodes for ease of representation. Protein clustering was performed using the Markov Cluster Algorithm (MCL) with inflation parameter of 1.4. Highlighted clusters were manually annotated.

**Venn diagrams.** To perform a GO-based analysis of protein content, categories of interest were manually defined from the list of protein hits contained in offspring GO terms of "Extracellular matrix (GO:0031012)", "Cell-adhesion (GO:0007155)", "Axon guidance (GO:0007411)" and "Axonogenesis (GO:0007409)" (see Supplementary Data 2). For each category, protein hits contained in each brain region were sorted according to their detection in mass spectrometry.

**PCA analysis.** For each brain target, Principal Component Analysis (PCA) was performed to highlight biological differences between injured and intact conditions across replicates. Proteins undetected in more than 5 samples were filtered out. Samples were plotted according to the first and second components, with the percentage of variation indicated for each component.

**Volcano plots.** For each brain target, protein hits were plotted according to the *p*-value and the $\log_2$ fold change in injured of differential expression between intact condition. Proteins with a FDR below 5% were highlighted.

**Heatmaps.** To analyse protein expression modulated by the injury (Supplementary Fig. 4), heatmaps were generated by plotting the difference between $\log_2$ normalized abundance of each protein hit and the mean across all samples. For representation, proteins were selected according their fold change between injured and intact conditions ($\log_2$ fold change >0.8) and the FDR-corrected p-value (FDR < 5% for all brain targets, except chiasm: FDR < 1% for ease of representation). To analyse protein expression of guidance proteins (Fig. 4), heatmaps were generated by plotting the normalized abundance across all replicates.

**Data representation.** Data analysis and representation were performed using R software for statistical computing (version 4.0.2).

## Western blot

For each brain target, 5–10 μg protein were loaded on SDS-PAGE gels (4–15% acrylamide, Biorad) and subjected to electrophoresis. Proteins were then transferred onto a nitrocellulose membrane (Thermo Fisher Scientific). Proteins were stained with Ponceau red to control for the transfer and the loading. Membranes were blocked with Tris-buffered saline 0.05% Tween (TBS-T) containing 5% milk for 1 h at room temperature and probed with primary antibody diluted in blocking solution overnight at 4 °C. The following primary antibodies and dilutions were used: anti-NCAM1 (1:1000, Rabbit, Cell signalling Technology, #99746), anti-Tenascin-C (1:1000, Rabbit, Abcam, ab108930), anti-Sema7A (1:1000, Rabbit, Abcam, ab23578), anti-Neuropilin-1 (1:1000, Rabbit, Cell signalling Technology, #3725), anti-Neuropilin-2 (1:1000, Rabbit, Cell signalling Technology, #3366), anti-Plexin-A4 (1:1000, Rabbit, Cell signalling Technology, #3816), anti-Ephrin-B3 (1:500, Rabbit, Invitrogen, 34-3600), anti-CSPG4 (1:1000, Rabbit, Proteintech, 55027-1-AP), anti-NrCAM (1:1000, Rabbit, Abcam, ab24344), anti-actin (1:5000, Mouse, Sigma-Aldrich, a1978). Membranes were washed in TBS-T and probed with horseradish peroxidase-conjugated secondary antibody (1:5000, anti-Rabbit, Proteintech; or 1:10000, anti-Mouse, ThermoFisher Scientific) for 1 h at room temperature. After several washed in TBS-T membranes were developed with ECL substrate (100 mM Tris-HCl pH 8.5, 0.5% coumaric acid, 0.5% luminol and 0.15% $H_2O_2$). Chemiluminescence signal was acquired using the ChemiDoc imaging system (Bio-Rad). The same membrane may be stripped and probed with different primary antibodies. Protein level was quantified with pixel densitometry and normalised to the level of actin. For each independent biological replicate, the pixel density of the protein of interest was normalized to the corresponding actin of the same samples. Data were normalized to intact condition. Bar graph data are represented as mean +/− standard error of the mean (SEM). Individual values are plotted on each graph. Data were subjected to two-tailed unpaired Student's *t*-test for statistical analysis, using GraphPad Prism version 9.1.2. For Western blot visual representation, one membrane probed with actin may appear several times to illustrate the quantification. Uncropped and unprocessed scans of the blots are supplied in the Source Data file.

## Transcriptomic datasets data screening

For the screen of gene expression in RGC, GEO datasets available on NCBI were used: atlas of neonatal (P5) RGC from single-cell transcriptomics analysis (accession number GSE115404[53]); atlas of adult RGC from single-cell transcriptomics analysis (accession number GSE137400[54]); microarray dataset comparing $PTEN^{-/-}$ $SOCS3^{-/-}$ RGC to WT RGC after optic nerve crush (accession number GSE32309[55]); RNA-sequencing dataset comparing Sox11-overexpressing RGC to Plap-overexpressing (control) RGC after optic nerve crush (accession

number GSE87046[56]). For single-cell transcriptomics analyses[53,54], online visualization tools were used to determine expression of genes of interest: https://health.uconn.edu/neuroregeneration-lab/rgc-subtypes-gene-browser/ and https://singlecell.broadinstitute.org/single_cell/study/SCP509/mouse-retinal-ganglion- cell-adult-atlas-and-optic-nerve-crush-time-series/. If the mean expression value was strictly positive, the gene was considered to be expressed in the corresponding cluster. For the microarray dataset[55], differential gene expression analysis was performed using the interactive web tool GEO2R (NCBI, https://www.ncbi.nlm.nih.gov/geo/info/geo2r.html) to plot the log fold-change and the FDR-corrected p-value. For the RNA-sequencing dataset[56], the complete gene list available on NCBI was used to plot the log fold-change and the FDR-corrected p-value.

## Intracardial perfusion

At the time of sacrifice, adult mice and mice at postnatal stages P4 to P14 were anaesthetized as described above, then intracardially perfused with ice-cold PBS for 3 min and with ice-cold 4% formaldehyde in PBS for 3 min. Brains were dissected out and samples were post-fixed overnight at 4 °C in 4% formaldehyde.

## Immunofluorescence on brain sections

Samples were post-fixed in 4% formaldehyde (Sigma) overnight and transferred to 30% sucrose for 2 days at 4 °C to cryoprotect. Samples were then embedded in tissue freezing medium compound (MM-France) and frozen at −80 °C. 30 μm and 20 μm thick coronal sections were performed for 10-week brain and young animals, respectively using a cryostat (CryoStar NX50, Thermo Fisher Scientific). 14 μm thick transversal sections were performed for optic chiasm. Immunohistochemistry on tissue sections was performed according to standard procedures. Sections were blocked for 1 h in PBS 0.1% Triton, 3% BSA, 5% Donkey Serum and incubated with the primary antibodies diluted in the blocking solution overnight at 4 °C. The following primary antibodies were used: anti-NCAM1 (1:100, Rabbit, Cell signalling Technology, #99746), anti-DCLK2 (1:100, Rabbit, Abcam, ab106639), anti-Sema4D (1:100, Rabbit, Abcam, ab134128), anti-CSPG4 (1:100, Rabbit, Proteintech, 55027-1-AP), anti-Ephrin-B3 (1:100, Mouse, R&D Systems, MAB395), anti-GFAP (1:200, Rat, Thermo Fisher Scientific, 13-0300), anti-Sema7A (1:100, Rabbit, Abcam, ab23578), anti-NrCAM (1:100, Rabbit, Abcam, ab24344), anti-Olig2 (1:100, Goat, R&D Systems, AF2418), anti-Iba1 (1:100, Goat, Novus Biologicals, NB100-1028), anti-NeuN (1:100, Mouse, Sigma-Aldrich, MAB377), anti-NeuN (1:100, Rabbit, Abcam, ab177487). Triton was omitted from the blocking solution for NCAM1, Sema4D, Sema7A and Ephrin-B3 immunostaining. For CSPG4, and Sema4D, heat-induced antigen retrieval was performed for 5 min in citrate buffer. After several washes by incubation with Alexa-fluor conjugated (anti-Rabbit, Thermo Fisher Scientific; anti-Mouse, Thermo Fisher Scientific; anti-Rat, Jackson Laboratory; anti-Goat, Thermo Fisher Scientific) antibodies according to standard protocol (dilution 1:200). Slides were mounted with Fluoromount-G with DAPI medium (Thermo Fisher Scientific).

## In utero eye injection

In utero eye injection was performed at E15.5 of timed pregnant WT female mice. Each pregnant female was anaesthetized and maintained under isoflurane using a facial mask during the whole time of surgery. The pregnant female was kept on a 37 °C plate during the whole time of surgery. After shaving and disinfecting the abdomen with 70% ethanol and betadine, laparotomy was performed using fine scissors in order to expose the uterus. The embryos were held in place through the uterus and frequently humidified with warm sterile PBS. In utero embryo eye injection of CTB-555 was done using a capillary connected to a pico-injector (Eppendorf). After injection of all embryos of the litter, the uterine horns were placed back into the abdominal cavity

and the abdominal wall and skins were stitched. Immediately after the surgery, the pregnant female was injected with 1X Buprecare (200 μl: 10 μl/g of animal weight; diluted in sterile PBS) and monitored regularly during recovery.

## Embryonic and postnatal sample collection

One day after in utero eye injection (at E16.5 of the timed pregnancy), each pregnant female was euthanized by cervical dislocation and the embryos were quickly removed from the uterine horns. Eye injection of CTB-555 was verified under a fluorescent binocular. For embryonic and postnatal P2 stages, the head of positive embryos was fixed in 4% formaldehyde overnight at 4 °C, then incubated in PBS 30% sucrose for cryopreservation. Heads and brains (from P4 and older) were embedded in tissue freezing medium and frozen at −80 °C until cryostat sectioning.

For Western Blot analysis, embryo heads were flash frozen on dry ice and thick slices were performed. Under a fluorescent binocular, the region of interest was microdissected. Samples were then frozen at −80 °C until protein extraction.

## In situ hybridization on retina sections

Templates of antisense in situ hybridization probes for EphB2 and EphA4 were cloned in a pGEMT easy vector (Promega) and synthesized using digoxigenin (DIG) RNA labelling Kit (Roche) after linearization of the plasmid. The following primers were used for template amplification by PCR from cDNA of mouse embryonic or adult brain: EphB2-forward: TCATAAGGGAAGTGACGGTTCT, EphB2-reverse: CCCTTGG TGTATTGCCTAAGTC; EphA4-forward: GGTATAAGGACAACTTCACG GC, EphA4-reverse: CTTCTGTGGTATAAACCGAGCC, Plexin-B1-forward: CCTCCGAGAGGCTCCAGATGCT, Plexin-B1-reverse: GCAGTGC CATCCTCCTCCAGG; Plexin-C1-forward: GGGACTTTCAAGCGACTG AG, Plexin-C1-reverse: AGTGTCTTGCGGAGATGCTT. In situ hybridization was performed as previously described[13]. Briefly, after hybridization of the DIG-labelled probe on slides, the alkaline phosphatase-conjugated anti-digoxigenin antibody (Roche) was incubated overnight at room temperature. Alkaline phosphatase staining was probed with NBT-BCIP (Roche) and slides were washed and post-fixed after the desired coloration intensity was obtained.

## Stripe assay

**Stripe set up.** Glass coverslips (diameter 1.2 mm) were coated with poly-L-lysine (0.5 mg/ml in ultrapure water, Sigma-Aldrich) and left overnight at room temperature. After two washes with ultrapure water, coverslips were dried.

Silicon matrices (Bastmeyer laboratory, Karlsruher Intitut für Technologie, matrix code 2B, channel width: 50 μm parallel) were boiled in ultrapure water, dried and then UV-treated. They were placed on the coverslips and first, recombinant mouse Ephrin-B3-FC chimera protein (10 μg/ml, R&D Systems, 7655-EB), recombinant mouse Sema4D-FC chimera protein (10 μg/ml, R&D Systems, 5235-S4B), recombinant mouse Sema7A-FC chimera protein (10 μg/ml, R&D Systems, 1835-S3), and Human IgG FC fragment (10 μg/ml, Millipore, 401104) as a control, were diluted in HBSS (Gibco, Fisher Scientific, 12082739) containing Alexa-Fluor 488 (1:500, Rabbit, Thermo Fisher Scientific) and injected in the chamber. After 30 min of incubation at 37 °C, matrixes were removed and two washes with PBS were performed. Human IgG FC fragment (10 μg/ml) was added (without fluorescent antibody) and incubated for 30 min at 37 °C. FC was then removed and two PBS washes were done before adding laminin (20 μg/ml) on coverslips for at least 2 h at room temperature. After two washes with ultrapure water, coverslips were covered with Neurobasal-A.

**Explant culture.** PTEN[fl/fl]SOCS[fl/fl] adult retina explants previously injected with AAV2-Cre and AAV2-CNTF were prepared according to

ref. 66. Briefly, mice were sacrificed by cervical dislocation following the institution's guidelines. Eyeballs were removed using Dumont's forceps #5 and dissected in ice-cold Hibernate A (BrainBits, HACA). Retina were dissected out and cut into small pieces (about 500 μm in diameter) with a scalpel. A retina explant was laid onto the coated coverslip before adding Neurobasal-A medium containing B-27 (Thermo Fisher Scientific), L-glutamine (Corning) and Penicillin-Streptomycin (Thermo Fisher Scientific).

**Immunofluorescence on explant culture.** After 7 days, explants were fixed and stained for β Tubulin III (TUJ1) (1:400, mouse, BioLegend, 801202) as described above, except that PBS was supplemented with 0.1% Triton for washing, primary antibody and secondary antibody steps.

**Data analysis.** Explants were imaged using an epifluorescence microscope. The number of axons in fluorescent and dark stripes were manually counted at the exit point of the explant. Each explant was considered as biologically independent. Regions of interest were designed on the sides of the explant orthogonal to the stripe direction. Regions of interest were discarded if the stripes did not have a sharp, clear pattern. The number of exiting neurites was normalized to the number of stripes for each region of interest. For each condition (Fc/Fc, Ephrin-B3-Fc/Fc, Sema4D-Fc/Fc, Sema7A-Fc/Fc), the number of neurites in fluorescent stripes and dark stripes were compared using two-tailed paired t-test, using GraphPad Prism version 9.1.2.

**Immunofluorescence on growth cones**
Adult retina explants from PTEN$^{fl/fl}$/SOCS3$^{fl/fl}$ mice injected with AAV2-Cre were obtained according to ref. 66. Briefly, glass coverslips were coated overnight with poly-L-lysine (0.5 mg/ml in ultrapure water, Sigma-Aldrich) then for 2 h with laminin (20 μg/ml, Sigma-Aldrich,). Adult retina explants were cultured on methylcellulose-containing coating medium. After 7 days of culture, explants were carefully fixed during 15 min at room temperature in PBS 4% formaldehyde 1.5% sucrose. After 3 washes of 10 min in 1X PBS, explants were incubated in primary antibody against EphB2 (1:200, Rabbit, Abcam, ab216629), EphA4 (1:200, Rabbit, Thermo Fisher Scientific, 21875-1-AP), Plexin-B1 (1:200, Mouse, R&D Systems, MAB3749), Plexin-C1 (1 :200, Mouse, R&D Systems, AF5375) and β-tubulin III (TUJ1, 1:500, Mouse, Biolegend, 801202 or Rabbit, Abcam, ab18207) overnight at 4 °C. After several washed in PBS, explants were incubated secondary antibodies for 2 h at room temperature (Alexa fluor 488-conjugated anti-Rabbit 1:400 (or 1:800 for Rabbit anti-Tuj1), Thermo Fisher Scientific; Alexa fluor 568-conjugated anti-Mouse, 1:400 (or 1:800 for Mouse anti-Tuj1), Thermo Fisher Scientific; Alexa fluor 647-conjugated Phalloidin, 1:400, Thermo Fisher Scientific). Coverslips were mounted with Fluoromount-G medium containing DAPI. Growth cones were imaged with Airyscan imaging and processing on a LSM710 confocal imaging unit (Zeiss) with the Zen software (version 2.1 SP3).

**Whole-mount tissue clarification and imaging**
**Brain transparization.** Whole-brain transparization was performed according to ref. 98. For visualization of the brain targets and optic tracts of the visual system, each eye of an adult WT mouse was injected with CTB-555 and CTB-647. After intracardial perfusion, brain and optic chiasm were dissected out and post-fixed in 4% formaldehyde. Brains were dehydrated in methanol, then bleached overnight in 6% H$_2$O$_2$ in methanol. After rehydration in PBS, brains were permeabilized in PBS 0.5% Triton for several days at 4 °C. The brain was dehydrated in methanol, then incubated for several hours in dichloromethane/methanol (2:1), then 30 min in dichloromethane (Sigma-Aldrich) before transparization in dibenzyl ether (Sigma-Aldrich). Transparized brain was imaged using the lightsheet microscope from LaVision

Biotec. Data processing and visualization were performed using Imaris software (x64 version 9.9.1).

**Optic nerve and chiasm clarification.** Optic nerve and chiasm clarification was performed according to ref. 66. After intracardial perfusion and post-fixation of the eyes, optic nerves attached to the optic chiasm were dissected and dehydrated in ethanol. Optic nerves were incubated for 2 h in hexane, then transparized in benzyl benzoate/benzyl alcohol (2:1) (Sigma-Aldrich). Optic nerves and chiasm were imaged using a spinning disk confocal microscope (Andor Dragonfly) with a custom stitching module (Metamorph version 7.10.2.240).

**Imaging**
Epifluorescence microscope Nikon Ti2 Eclipse was used for imaging of brain sections and co-cultures, with NIS-Elements software (version 5.11.02). Cleared brain was imaged using lightsheet microscope from LaVision Biotec. Cleared optic nerves, optic chiasm, and brain slices labelled with cell markers were imaged using a spinning disk confocal microscope (Andor Dragonfly spinning disk confocal microscope, Oxford Instruments). Images were analysed using Fiji software (version 2.0.0).

**Regeneration and axon guidance quantification**
For axon regeneration in wild-type condition, individual CTB-positive axons were manually counted at defined distances from the lesion site. For axon regeneration in the long-distance regeneration model, individual CTB-positive axons were manually counted in the ipsilateral side along a line at 100 μm before the optic chiasm. For axon guidance in the long-distance regeneration model, CTB-positive axons were manually counted along a line at 200 μm into the optic chiasm, and along the midline. Bar graph data are represented as mean + /− standard error of the mean (SEM). Individual values are plotted on each graph. For each shRNA against guidance receptors (sh EphA4+sh EphB2 and sh Plexin-B1), the same dataset of sh scrambled was used to compare the number of axons. The number of axons were compared using two-tailed unpaired Student's t-test, using GraphPad Prism version 9.1.2.

**Reporting summary**
Further information on research design is available in the Nature Research Reporting Summary linked to this article.

## Data availability

The LC-MS/MS data have been submitted to the ProteomeXchange Consortium via the PRIDE[99] partner repository under dataset identifier PXD029325. For peptide and protein identification with Mascot, concomitant searches were done against Uniprot database [https://www.uniprot.org/] (Mus Musculus taxonomy, for Optic chiasm and SCN samples, downloaded in June 2019 with 87573 entries, and for dlGN, vLGN and SCol samples, downloaded in August 2020 with 87975 entries). The protein expression data generated in this study are available in Supplementary Data 1–4. The Western blot data generated in this study are provided in the Source Data file. Quantification and statistics of source data are provided in the Source Data file. Source data are provided with this paper. RGC gene expression data used in this study are available on NCBI's GEO database under the accession codes: GSE115404 (atlas of neonatal (P5) RGC from single-cell transcriptomics analysis, user-friendly version at https://health.uconn.edu/neuroregeneration-lab/rgc-subtypes-gene-browser/); GSE137400 (atlas of adult RGC from single-cell transcriptomics analysis, user-friendly version at https://singlecell.broadinstitute.org/single_cell/study/SCP509/mouse-retinal-ganglion-cell-adult-atlas-and-optic-nerve-crush-time-series/); GSE32309 (microarray dataset comparing PTEN$^{-/-}$ SOCS3$^{-/-}$ RGC to WT RGC after optic nerve crush); GSE87046 (RNA-sequencing dataset comparing Sox11-overexpressing RGC to

control RGC after optic nerve crush). Source data are provided with this paper.

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

## Acknowledgements

We thank Dr Kim Ba-Charvet and Dr Charlotte Decourt for precious advice, and Dr Julien Courchet (INMG, Lyon, France) and Dr Xavier Nicol (Institut de la Vision, Paris, France) for their comments and feedbacks during Noemie Vilallongue thesis committee meetings. We thank the Grenoble Institute of Neuroscience animal facility. This work was supported by the Photonic Imaging Center of Grenoble Institute Neuroscience (Univ Grenoble Alpes – Inserm U1216) which is part of the ISdV core facility and certified by the IBiSA label. The proteomic experiments were partially supported by Agence Nationale de la Recherche under projects ProFI (Proteomics French Infrastructure, ANR-10-INBS-08) and GRAL, a program from the Chemistry Biology Health (CBH) Graduate School of University Grenoble Alpes (ANR-17-EURE-0003). This work was supported by a grant from ANR to HN (C7H-ANR16C49) and SB (ANR-18-CE16-0007). HN is supported by NRJ Foundation and the European Research Council (ERC-St17-759089). JS is supported by Fondation pour la Recherche Médicale (FRM) postdoctoral fellowship (SPF201909009106). CD is supported by a grant from the French National Research Agency in the framework of the "investissements d'Avenir" program (ANR-15-IDEX-02 NeuroCoG) and by Fondation de France, Berthe Fouassier grant.

## Author contributions

H.N. and S.B. conceptualized the study. N.V., J.S., C.D., B.B., A.P., E.P., and B.E. performed experiments. Y.C. and A.M.H. performed mass spectrometry-based proteomic data collection and analysis. N.V. and J.S. analyzed and interpreted data. N.V. and J.S. drafted figures. N.V., J.S., S.B., and H.N. wrote the manuscript.

## Competing interests

The authors declare no competing interests.
