## [Peer Review File · Nature Communications]

REVIEWER COMMENTS

Reviewer #1 (Remarks to the Author):

This is an important work but needs improvements to bolster its potential impact. The authors performed mass-spectrometry based proteomics to uncover the proteomes of major nuclei of the adult mouse visual system. They demonstrate that expression of specific proteins is modulated by optic nerve crush, compared to uninjured. Consistent with previous reports, the authors show that one specific protein, EphB2, functions as a deterrence cue for RGC axons in vitro. However, they do not demonstrate that any other proteins identified by their proteomics analysis function as guidance cues. There is no attempt to validate the function of candidate guidance molecules identified for their role in RGC axon growth or regeneration.

Specific Issues:

1) Authors argue that microdissection and proteomic analysis of intermediate and terminal RGC axon target sites can be utilized to infer extracellular signals that may influence guidance in the adult brain. However, these experiments do not experimentally distinguish between intracellular peptides, cell-surface peptides, and extracellular secreted peptides. It is difficult to conclude based on the evidence provided which of the molecules identified by these proteomics experiments are secreted factors for guidance of RGC axons.

2) The authors do not validate candidate peptides as being required for RGC axon growth or regeneration neither in culture or in vivo. In figure 7 the authors demonstrate RGC explants are deterred from growth towards EphB2 expressing COS cells. This is consistent with previous studies and is a useful control. However, the authors do not demonstrate whether any novel factors identified by proteomics analysis perturb or promote axon growth.

3) Fig 4 I & J: should include quantification of fluorescence, with error bars. One image shown is not sufficient to demonstrate consistent differences in expression for Col6a3 between post-crush and control.

Reviewer #2 (Remarks to the Author):

Nawabi

In a previous paper of which they were co-first authors, Belin, Nawabi and colleagues used a proteomic method to analyze effects of axotomy on retinal ganglion cells (RGCs) (Belin et al., 2015). Here they use a similar method to profile four target areas of RGCs – dorsal and ventral lateral geniculate (dLGN, vLGN), superior colliculus and suprachiasmatic nucleus – as well as an intermediate target, the optic chiasm. Their focus is on the presence of proteins involved in axon guidance. In brief, Figure 1 shows the method, Figures 2 and 3 summarize proteomes of normal adult structures, Figures 4 and 5 do the same following axotomy, and Figure 6 shows a developmental series for one target, the dLGN. (I discuss Figure 7 below).

The results are convincing, straightforward, and well-presented. The drawback is that the analysis is more anecdotal than systematic. The authors focus on proteins involved in axon guidance, which is reasonable, but one ends up with little more than lists of adhesion molecules, matrix molecules, glial molecules and intracellular molecules that might be involved. There is little consideration of which might be important. For molecules in target areas, it is not even clear whether they would play roles in guidance, which is usually viewed as being mediated by molecules on the way to the target rather than (or in addition to) in the target. There is no consideration of what cells within the target areas produce or accumulate the proteins – in other words, their cellular localization. Changes following axotomy are difficult to interpret, in that they presumably result from loss of molecules on axons as well as expression changes in the target. This is particularly problematic for the chiasm. In addition, RGC axons provide a minority of inputs to most of these areas, so they cannot be said to be deafferented. The developmental analysis suffers from similar problems, as well as issues related to how this very small structure was isolated at early stages. In summary, lack of analysis renders this paper a catalogue that will be useful as a resource if presented in an archival journal, but not one that provides novel insights.

It is also odd that the vLGN and dLGN are lumped together for analysis in that differences in their target properties have been documented (e.g. Su, *J Neurosci.* 2011; Sabbagh, *J Neurochem*, 2018). It is also unfortunate that the unbiased proteomic analysis of Sabbagh is not cited.

Figure 7 presents experiments showing that RGC axons growing from explants are repelled by Ephrin-B3. Although it is true that this particular genotype has not been assayed, the general result has been reported previously. The description also fails to take account of several important prior results. For example: (a) EphrinB is thought to selectively repel ipsilaterally projecting RGCs, which constitute a small minority (Williams, *Neuron*, 2003). This seems at odds with the broad effect seen here. (b) Injured RGCs upregulate Ephrin-B3 (Liu, *J Neurosci*, 2006), making interpretation of their responses to exogenous Ephrin difficult. (c) Further complicating this issue, astrocytes in the optic nerve head upregulate Ephrin-B3 following axotomy (Fu, *IOVS*, 2010). Finally, the specificity of the effect is not documented here. In short, this preliminary result does not add substantially to the paper.

Reviewer #3 (Remarks to the Author):

To address the yet unsolved problem of insufficient re-innervation and functional recovery of nerves in the central nervous system after injury, Vilallongue and colleagues generated and analyzed a rich proteomic data set for regenerative axons. The work is following a clear rationale and is definitely technically sound. The authors analyzed the proteomes of four different regions in the murine brain, which are targeted by retinal ganglion cell axons. Interestingly, these regions show clear changes in their proteomes after bilateral optic nerve injury. By functional ex-vivo assays, the work shows that RGC axons are still responsive to guidance cues. The proteomically defined guidance landscape provided by the authors is supporting the hypothesis that inefficient re-innervation is caused by miss-guidance rather than a guidance block. Thus, the data set clearly provides novel insights into the mechanisms of neurite regrowth in response to an injury and may serve as a sound data foundation for future functional studies. The conclusions drawn are convincing as they are clearly justified by the experimental data provided. The data are documented according to the standards of the field satisfying the MIAPE (Minimum Information About a Proteomics Experiment) standards for quantitative proteomic data.

I just have minor comments:

(1) Page 24/ line 530: As recommended by the MIAPE standards, the number of entries (Mus musculus taxonomy) should be given along with the version of the database used for protein/peptide identification.

(2) The raw data have been uploaded to PRIDE. I would recommend to provide an experimental template allowing to assign single runs to the different conditions. In addition, it would be good to provide a more detailed description of the experimental parameters there.

(3) The figures are of sufficient quality, comprehensively summarizing the outcome of the data analysis. Nevertheless, the fonts are very small at a quite low resolution making it sometimes difficult to read the labels. The authors might consider to increase the resolution and/or font size.

In conclusion, given its significance for the field as well its sound technical quality, I can recommend the manuscript for its publication in Nature Communications with minor revisions.

Manuscript “Guidance landscapes unveiled by quantitative proteomics to control reinnervation in adult visual system”, Vilallongue et al.

Response to the Reviewers (in blue below)

Reviewer #1 (Remarks to the Author):

This is an important work but needs improvements to bolster its potential impact. The authors performed mass-spectrometry based proteomics to uncover the proteomes of major nuclei of the adult mouse visual system. They demonstrate that expression of specific proteins is modulated by optic nerve crush, compared to uninjured. Consistent with previous reports, the authors show that one specific protein, EphB2, functions as a deterrence cue for RGC axons in vitro. However, they do not demonstrate that any other proteins identified by their proteomics analysis function as guidance cues. There is no attempt to validate the function of candidate guidance molecules identified for their role in RGC axon growth or regeneration.

We thank the Reviewer for his/her interest in our work and for his/her precious advice that helped us improve our manuscript. In this revised version of our manuscript, we addressed all the raised issues. Our point-by-point answer is detailed hereafter, and also added in the answer to specific issues (see below).

To address specifically your comment on functional follow-up studies, we used our map of guidance molecules in the mature CNS that we have generated (initially Figure 8, now Figure 6). From this map, we chose 3 pairs of guidance ligand/receptor to be challenged ex vivo and in vivo, namely: Sema4D/PlexinB1, Ephrin-B3/EphA4-EphB2 and Sema7A/PlexinC1 (as a control). We show that their modulation in wild-type animals improves axon regeneration in the optic nerve (Supplementary Figure 7). Strikingly, when combined with a model of long-distance regeneration, modulation of the guidance signaling allows regenerating axons to enter the chiasm and to cross the midline.

In more details, in this updated version of the manuscript, we demonstrate that axon guidance is a key process to control the navigation of regenerating axons. To do so, we used the map of guidance cues expression (initially Figure 8, now Figure 6) and focused on two relevant guidance cues expressed in the chiasm: Sema4D and Ephrin-B3. We also tested Sema7A that is not expressed at the chiasm. We verified the proteomics results by immunostaining on chiasm sections (updated Figure 7A). We showed that Sema4D and Ephrin-B3 are expressed in the optic chiasm, but not Sema7A, consistently with the proteomic dataset.

Next, we set up a new ex vivo technique (updated Figure 7), by combining our mature retina explant culture assay with a stripe assay. This assay shows the preferential growth of axons on different substrates or guidance cues laid on alternating stripes (Walter et al., 1987). Interestingly, we found that newly growing axons are repelled by Ephrin-B3 and Sema4D, respectively. In contrast, they are not responsive to Sema7A. This result shows that regenerating axons are able to respond to guidance cues and that their response is specific to certain guidance molecules.

Based on these exciting results, we modulated Sema4D and Ephrin-B3 guidance signaling in vivo. We did so by silencing their corresponding receptors, PlexinB1 and EphA4-EphB2 respectively, with shRNA-expressing vectors specifically targeting retinal ganglion cells (updated Figure 8). In wild-type animals, this modulation improves axon regeneration in the optic nerve. Strikingly, when Sema4D/PlexinB1 or Ephrin-B3/EphA4-EphB2 signaling is silenced in combination with a system primed for regeneration (upon mTOR and JAK/STAT activation in retinal ganglion cells), guidance of regenerating axons is modified at the optic chiasm. More precisely, when silencing Sema4D or Ephrin-B3 guidance signaling, regenerating axons are able to enter the chiasm and to cross the midline. Our results show that axon guidance is key to control the trajectory of regenerating axons, a process that is highly relevant to functional regeneration in vivo.

Specific Issues:

1) Authors argue that microdissection and proteomic analysis of intermediate and terminal RGC axon target sites can be utilized to infer extracellular signals that may influence guidance in the adult brain. However, these experiments do not experimentally distinguish between intracellular peptides, cell-surface peptides, and extracellular secreted peptides. It is difficult to conclude based on the evidence provided which of the molecules identified by these proteomics experiments are secreted factors for guidance of RGC axons.

We thank the Reviewer for raising this point. The scope of the paper is to provide a map of guidance cues expressed on the putative path of regenerating axons, with the ultimate goal of guiding regenerating axons toward their targets. This is the reason why we focused on the primary visual targets of retinal ganglion cells (SCN, vLGN, dLGN and SCol), as well as the particular choice-point that is the optic chiasm. This unbiased approach allowed us to identify a number of guidance and growth-related cues expressed in each visual target. We classified these cues according to their known contribution to axon growth during development, which includes categories such as extracellular molecules, adhesion molecules, guidance factors, etc. (updated Figure 2 and Supplementary Figure 2).

To answer the Reviewer's point, we now provide additional information regarding the sub/extra-cellular localization of such cues with immunofluorescence and confocal imaging (updated Figure 2 and Supplementary Figure 2). This allows us to determine that some cues, such as CSPG4 and Ncam1, are expressed at the cell surface of NeuN-positive cells, possibly associated with perineural nets (PNN, as described by Sabbagh et al., 2018). Other cues, such as Sema4D, are expressed by Olig2-positive oligodendrocytes, consistently with Moreau-Fauvraque et al., 2006. Others cues are expressed by astrocytes, such as NCAM. These results support that guidance-related cues expressed in the mature brain may play different roles (eg axon guidance or fasciculation) depending on the brain region.

2) The authors do not validate candidate peptides as being required for RGC axon growth or regeneration neither in culture or in vivo. In figure 7 the authors demonstrate RGC explants are deterred from growth towards EphB2 expressing COS cells. This is consistent with previous studies and is a useful control. However, the authors do not demonstrate whether any novel factors identified by proteomics analysis perturb or promote axon growth.

To answer the Reviewer's point regarding the involvement of candidate proteins in RGC axon growth or regeneration, we used the map of guidance cues expression (now Figure 6) and chose two relevant guidance cues expressed in the chiasm: Sema4D and Ephrin-B3. We also tested Sema7A that is not expressed in the chiasm. We verified the proteomics results by immunostaining on chiasm sections (updated Figure 7a). Consistently with the proteomic dataset, we showed that Sema4D and Ephrin-B3 are expressed at the optic chiasm, but not Sema7A.

Next, we set up a new *ex vivo* technique (updated Figure 7), by combining our mature retina explant culture assay with a stripe assay. This assay shows the preferential growth of axons on different substrates or guidance cues layed on alternating stripes (Walter et al., 1987). Interestingly, we found that newly growing axons are repelled by Ephrin-B3 and Sema4D, respectively. In contrast, they are not responsive to Sema7A. This result shows that regenerating axons are able to respond to guidance cues and that their response is specific to certain guidance molecules.

Based on these compelling results, we modulated Sema4D and Ephrin-B3 guidance signaling *in vivo*. We did so by silencing their corresponding receptors, PlexinB1 and EphA4-EphB2 respectively, with shRNA-expressing vectors specifically targeting retinal ganglion cells (updated Figure 8). In wild-type animals, this modulation improves axon regeneration in the optic nerve. Strikingly, when Sema4D/PlexinB1 or Ephrin-B3/EphA4-EphB2 signaling is silenced in combination with a system primed for regeneration (upon mTOR and JAK/STAT activation in retinal ganglion cells), guidance of regenerating axons is modified at the optic chiasm. More precisely, when silencing Sema4D or Ephrin-B3 guidance signaling, regenerating axons are able to enter the chiasm and to cross the midline. Our results show that axon guidance is key to control the trajectory of regenerating axons, a process that is highly relevant to functional regeneration *in vivo*.

In summary, in the present study, our point is to show that axon guidance, quasi-exclusively studied during development so far, is in fact at play in the mature CNS. The novelty lies in three facts: i) guidance cues are expressed in the adult central nervous system; ii) their expression is not modified upon injury; and iii) adult regenerating axons are able to respond to these guidance cues. These elements are added to the Discussion section.

3) Fig 4 I & J: should include quantification of fluorescence, with error bars. One image shown is not sufficient to demonstrate consistent differences in expression for Col6a3 between post-crush and control.

We thank the Reviewer for pointing this oversight. We now provide a Western blot-based analysis of protein expression of other candidate molecules in intact versus injured condition. We analyzed the expression of the extracellular matrix protein Tenascin-C as an example. Consistent with the proteomic data, Tenascin-C shows no change of expression in the chiasm, while being significantly downregulated upon injury in the SCN. This was done on three different animals, with pixel densitometry quantification and unpaired t-test statistical analysis. As a second example, we analyzed GFAP expression in the SCN and in the chiasm, using

immunofluorescence. We quantified fluorescence intensity on cryosections obtained from three different animals, and performed an unpaired t-test statistical analysis, as detailed in the Material and Methods. Consistently with the differential expression analysis, we show that GFAP remains stable in the SCN while being upregulated in the optic chiasm after injury. These results are in updated Figure 3.

Reviewer #2 (Remarks to the Author):Nawabi

In a previous paper of which they were co-first authors, Belin, Nawabi and colleagues used a proteomic method to analyze effects of axotomy on retinal ganglion cells (RGCs) (Belin et al., 2015). Here they use a similar method to profile four target areas of RGCs – dorsal and ventral lateral geniculate (dLGN, vLGN), superior colliculus and suprachiasmatic nucleus – as well as an intermediate target, the optic chiasm. Their focus is on the presence of proteins involved in axon guidance. In brief, Figure 1 shows the method, Figures 2 and 3 summarize proteomes of normal adult structures, Figures 4 and 5 do the same following axotomy, and Figure 6 shows a developmental series for one target, the dLGN. (I discuss Figure 7 below).

The results are convincing, straightforward, and well-presented. The drawback is that the analysis is more anecdotal than systematic. The authors focus on proteins involved in axon guidance, which is reasonable, but one ends up with little more than lists of adhesion molecules, matrix molecules, glial molecules and intracellular molecules that might be involved. There is little consideration of which might be important.

We thank the Reviewer for his/her interest in our study. We would like to thank him/her for all comments that helped us improve our manuscript and clarify our point.

Despite major advances in adult neuroregeneration in the past decade, functional circuit has not been obtained yet. For a large part, this is due to axons failing to resume the expected navigation pattern that would drive them to their correct target (Belin et al., 2015; Lim et al., 2016). Hence, too few regenerating axons reach their functional target, and/or axons connect to other areas, leading to irrelevant/non-functional circuit connections.

Over the past decades, much work has been done to produce these maps during development - including the article mentioned, Sabbagh et al., 2018, as well as work from T. Jessell's, M. Tessier Lavigne's and C. Mason's labs among others. In the context of adult regeneration, however, little is known about the guidance landscape in the mature central nervous system. Second, to our knowledge, nothing is known about the possibility to guide adult regenerating axons towards their correct target. To address these two issues, we designed a broad proteomics-based analysis of the primary visual targets of retinal ganglion cells (SCN, LGN, SCoI) and of the particular choice-point that is the optic chiasm.

This analysis allowed us to draw a comprehensive map of guidance cues expression in the mature visual system (updated Figure 6). This map is of crucial importance to address guidance defects observed in regenerative models. Indeed, resuming a

functional circuit in adult highly relies on the characterization of the environment. In this context, while the lesion site has been extensively studied in the past, nothing is known about molecular cues expressed in distal regions, a fortiori in the target regions of the brain.

To demonstrate the importance of our analysis, we used our map of guidance molecules in the mature CNS that we generated using mass spectrometry approach (initially Figure 8, now Figure 6). From this map, we chose three pairs of guidance ligand/receptor to be challenged ex vivo and in vivo, namely: Sema4D/Plexin-B1, Ephrin-B3/EphA4-EphB2 and Sema7A/Plexin-C1 (as a control). We show that Sema4D and Ephrin-B3 are expressed at the optic chiasm but not Sema7A (as revealed by the proteomic dataset, updated Figure 7). Using an optimised ex vivo approach, we show that adult regenerating axons respond to Sema4D and EphrinB3, while remaining insensitive to Sema7A (updated Figure 7). Modulation of the guidance signaling in wild-type animals improves axon regeneration in the optic nerve (Supplementary Figure 7). Strikingly, when Sema4D/Plexin-B1 or Ephrin-B3/EphA4-EphB2 modulation is combined with a system primed for regeneration (upon mTOR and JAK/STAT activation in retinal ganglion cells), axon guidance at the optic chiasm is modified (updated Figure 8). More precisely, when silencing Sema4D or Ephrin-B3 guidance signaling, regenerating axons are able to enter the chiasm and to cross the midline. Our results show that axon guidance is key to control the trajectory of regenerating axons, a process that is highly relevant to functional regeneration in vivo.

We agree with the Reviewer, that our presentation was not ideally on point, so we rephrased the Results section to focus specifically on the axon guidance aspects of our analysis (text corresponding to updated Figures 2-4).

For molecules in target areas, it is not even clear whether they would play roles in guidance, which is usually viewed as being mediated by molecules on the way to the target rather than (or in addition to) in the target.

We thank the Reviewer for his/her meaningful comment. Different studies showing long-distance regeneration highlighted that guidance defects are present not only in the vicinity of RGC brain targets, but also along the trajectory. As during development, a critical intermediate target is the optic chiasm. Indeed, several models of axon regeneration (Belin et al., 2015; Lim et al., 2016) revealed many guidance defects in the optic chiasm. Therefore, in this manuscript, we chose to focus on the optic chiasm to address the effect of guidance signaling on the trajectory of regenerating axons. We tested two guidance cues – namely Sema4D and Ephrin-B3 - that are expressed in the optic chiasm and we showed as a proof-of-concept that modulation of these signalings controls axon pathfinding (updated Figure 8 and Supplementary Figure 7). The next step, which is beyond the scope of this work, would be to study brain target innervation and the functionality of the circuit.

There is no consideration of what cells within the target areas produce or accumulate the proteins – in other words, their cellular localization.

We now provide confocal imaging of guidance cues with co-labeling of central nervous system cell markers (updated Figure 2 and Supplementary Figure 2). Most of the guidance cues are associated with neurons, as shown for NCAM1 and CSPG4. Previous studies addressing proteomic changes in the developing LGN show that these molecules are associated with neurons and more precisely with perineural nets (PNN) (Sabbagh et al., 2018). In our case, the pattern of expression makes it highly possible that guidance cues are associated with PNN. Interestingly, we found that Sema4D expression associates with oligodendrocytes, as reported previously (Moreau Fauvraque et al., 2003; Zhang et al., 2014). Even if we found Iba-positive microglia cells in the mature brain, they do not express any of the guidance cues we studied. Moreover, depending on the brain regions some cues are associated with one or another cell type. For example, NCAM1 is associated with GFAP-positive astrocytes in the chiasm, whereas its expression is associated with neurons in the SCN, LGN and SCol (updated Figure 2).

Together, these results suggest that guidance cues play different roles depending on the brain region and on the cellular subtype. For example, CAM have been shown to regulate axon fasciculation when expressed by neurons (Kamiguchi et al., 2007). Interestingly, during development, growth cones from pioneer axons interact with glia cells at critical choice points (Rigby et al., 2020) and CAM, specifically NCAM stands as one of the major effectors during this navigation (Neugebauer et al., 1988). Thus, the expression of NCAM1 by astrocytes in the mature optic chiasm may be crucial for axon navigation during regeneration, particularly for midline crossing. Beside their role in cell adhesion, CAMs are also been involved in modulating axonal response to guidance cues (Castellani et al., 2000; Nawabi et al., 2010). We have added all these elements in the Discussion section (lines 425-442, 489-500).

Changes following axotomy are difficult to interpret, in that they presumably result from loss of molecules on axons as well as expression changes in the target. This is particularly problematic for the chiasm.

We thank the Reviewer for his/her comment. The goal of this study is primary to address the guidance landscape of mature visual system and how optic nerve injury affects the expression of guidance cues. Indeed, all regenerative models display axon guidance defects that may contribute to the failure of functional recovery. Yet, the guidance landscape in the mature brain is elusive.

Interestingly, we found that the guidance maps are not affected by optic nerve injury (updated Figure 4) and that these maps are established during development of the visual circuit (updated Figure 5). In addition, our proteomic data show that the injury affects the proteome of visual targets (updated Figure 3), including via an upregulation of inflammatory pathways upon injury, as supported by a recent transcriptomic profiling (Kaiser et al., 2019). We also found a modulation of proteins involved in circuit maintenance and synaptic functions. From our study, we found that the brain regions distal to the injury site can integrate injury signals and respond to them. These proteomic changes may indirectly involved in guidance, as well as other processes such as synapse formation, maintenance and pruning. We have added elements of discussion in the relevant section.

In addition, we now provide new results showing which cell types express guidance cues of interest (updated Figure 2 and Supplementary Figure 2).

In addition, RGC axons provide a minority of inputs to most of these areas, so they cannot be said to be deafferented.

We agree with the Reviewer that we overstated our purpose and that it is incorrect to phrase « deinnervation » of the visual targets. We corrected this point in our manuscript by focusing on the loss of retina inputs within these brain regions.

The developmental analysis suffers from similar problems, as well as issues related to how this very small structure was isolated at early stages.

Based on the differential expression analysis of intact versus injured conditions, we found that guidance factors are unchanged upon optic nerve injury, which correlates with the failure of reinnervation of visual targets by regenerating axons. We then sought to determine when this guidance landscape is established. To illustrate our analysis, we focused on the LGN, whose development, innervation and maturation are well documented. We selected several guidance molecules that belong to each family of canonical guidance factors and that were identified in Figure 2A. We then studied their expression at key timepoints of LGN development. In this context, the analysis of developmental stages supports the idea that guidance factors are dynamically regulated during circuit formation – particularly during innervation of the functional targets.

The LGN was identified and isolated thanks to injection of Alexa555-conjugated CTB in eyes 6 to 24 hours prior to dissection at developmental stages of interest. We now provide more details on the microdissection procedure in the Material and Methods section (lines 794-803). Using immunofluorescence and Western blot analysis, our results reveal that the pattern of expression of families of guidance cues is dynamically regulated during maturation of the LGN. This pattern is maintained throughout adulthood and remains steady upon injury (updated Figure 5 and Supplementary Figure 5).

In summary, lack of analysis renders this paper a catalogue that will be useful as a resource if presented in an archival journal, but not one that provides novel insights.

In the context of CNS regeneration, adult axon guidance has never been addressed so far. The expression of guidance cues in the mature brain and their regulation upon injury (distal to the lesion site) have also never been characterized in an unbiased manner. In this view, our work is of crucial importance and goes beyond a mere catalogue, as we provide:

- 1- A comprehensive map of guidance and guidance-associated factors expressed in the mature visual system;
- 2- A functional proof-of-concept that the trajectory of regenerating axons can be controlled *ex vivo* and *in vivo*, based on this map and on modulation of specific guidance signaling.

We acknowledge that the initial version of our manuscript may not have stressed out these points, and we hope to clarify our purpose with the following follow-up studies.

To answer his/her concerns, we updated the text in the Results and Discussion sections, provided new results and explained in detail why this work is not a catalogue, but a functional guidance map highly relevant to address pathfinding of regenerating axons in the mature brain. In particular, the new concept of axon guidance in adult regeneration is supported by two pairs of guidance ligand/receptor, whose modulation strongly impacts axon pathfinding in vivo in the mature brain.

In more details, as mentioned above, we used our map of guidance molecules in the mature CNS that we generated (initially Figure 8, now Figure 6). From this map, we chose three pairs of guidance ligand/receptor to be challenged *ex vivo* and *in vivo*, namely: *Sema4D/Plexin-B1*, *Ephrin-B3/EphA4-EphB2* and *Sema7A/PlexinC1* (as a control). We show that *Sema4D* and *Ephrin-B3* are expressed at the optic chiasm but not *Sema7A* (as revealed by the proteomic dataset, updated Figure 7).

Using an optimised *ex vivo* approach, we show that adult regenerating axons respond to *Sema4D* and *EphrinB3*, while remaining insensitive to *Sema7A* (updated Figure 7). Modulation of the guidance signaling in wild-type animals improves axon regeneration in the optic nerve (Supplementary Figure 7).

Strikingly, when *Sema4D/PlexinB1* or *Ephrin-B3/EphA4-EphB2* modulation is combined with a system primed for regeneration (upon mTOR and JAK/STAT activation in retinal ganglion cells), axon guidance at the optic chiasm is modified (updated Figure 8). Indeed, we show that silencing of *Sema4D* or *EphrinB3* signalling causes axons to enter the chiasm and to cross the midline.

Altogether our results demonstrate i) the relevance of our proteomics approach and of the resulting guidance map in the mature brain; ii) that mature axons are able to respond to axon guidance signals; and iii) more importantly, that axon guidance is critical for proper regeneration and circuit formation after injury, with the identification and validation of two key pairs of ligand/receptor. Our updated manuscript provide the first *in vivo* proof-of-concept of such processes in the mature CNS. Our findings will be key for future strategies aimed at building functional circuits during regeneration.

It is also odd that the vLGN and dLGN are lumped together for analysis in that differences in their target properties have been documented (e.g. Su, *J Neurosci.* 2011; Sabbagh, *J Neurochem*, 2018). It is also unfortunate that the unbiased proteomic analysis of Sabbagh is not cited.

In updated Supplementary Figure 2, we have updated the representations and now provide 5 set-Venn diagrams to keep each target separate. We have updated Table 3 to provide information for each target. We have also kept all targets separate in the representation of updated Figure 6. Our proteomic analysis does not allow us to formally compare the visual targets between each other. Only the quantitative comparison between intact and injured conditions was performed. Nevertheless, our immunofluorescence experiment (updated Figure 2), allows us to validate expression of guidance molecules and associated factors, in particular the ECM molecule CSPG4. Our results are fully consistent with the work of Sabbagh et al., 2018, that we now add in the references. We thank the Reviewer for pointing the literature oversight. We corrected this mistake by adding the corresponding references in the Results and Discussion sections.

Figure 7 presents experiments showing that RGC axons growing from explants are repelled by Ephrin-B3. Although it is true that this particular genotype has not been assayed, the general result has been reported previously.

We thank the Reviewer for his/her careful criticism of this result. The purpose of our study is to demonstrate that axon guidance contributes to the pattern of regenerating axons in the lesioned mature brain. Our study shows for the first time that mature axons, in a context of regeneration, can integrate guidance signals and respond to them to orient their navigation.

For this purpose, we used an assay optimised by our lab : the culture of adult RGC explants. We now combine this with a stripe assay (preferential growth assay) to demonstrate that adult RGC axons are repelled by two guidance cues : Ephrin-B3 and Sema4D. In parallel, we give evidence that the corresponding canonical receptors of these guidance factors are expressed by RGC, in particular on their growth cones (updated Figure 7).

In a second step, we take this result forward with a shRNA-based receptor-silencing approach combined with a long-distance regeneration model (updated Figure 8). This in vivo experiment provides evidence that the modulation of guidance signaling can modify the trajectory of growing axons in the context of adult regeneration. This has never been shown before and represents a strong proof-of-concept in the field of axon regeneration and circuit repair. Altogether, our experiments are designed to demonstrate the possibility and the relevance to guide regenerating axons towards their target.

The description also fails to take account of several important prior results. For example: (a) EphrinB is thought to selectively repel ipsilaterally projecting RGCs, which constitute a small minority (Williams, Neuron, 2003). This seems at odds with the broad effect seen here. (b) Injured RGCs upregulate Ephrin-B3 Liu, J Neurosci, 2006), making interpretation of their responses to exogenous Ephrin difficult. (c) Further complicating this issue, astrocytes in the optic nerve head upregulate Ephrin-B3 following axotomy (Fu, IOVS, 2010).

We thank the Reviewer for his/her excellent suggestions to discuss what is known about Ephrin-B signaling and RGC axon growth.

(a) First, the selectivity of EphrinB1 signaling for lateral projections at the optic chiasm has been demonstrated during development. The fact that adult axons respond to guidance cues does not necessarily mean that they have the exact same properties as embryonic neurons, notably regarding the spatio-temporal regulation of the effect. So, it is well possible that the repulsive effect observed using our adult assay affects the majority of RGC axons, with no segregation of ipsi- versus contralateral projections. Moreover, the possibility that the various RGC subpopulations differ in their guidance response further adds to the complexity of the characterization of axon guidance in adult. These elements of primary importance are beyond the scope of our present study, and we now provide elements of discussion in the corresponding section (lines 510-516).

(b) For Ephrin-B3 and Sema4D signaling, we show that expression of the corresponding receptors, EphA4-EphB2 and Plexin-B1 respectively, is maintained in the regeneration model after injury, in most of the RGC (Figure 7). As for expression of guidance cues by RGC, such as Ephrin-B3 itself and others presented in Supplementary Figure 6, their expression can be modulated within RGC after axon injury. This would be very important to consider in the context of regeneration, as the interplay of Ephrin-B signaling may play a role in the guidance response of adult axons. Here, we provide ex vivo and in vivo proof-of-concepts, based on two different guidance signaling, that adult guidance is a crucial phenomenon for axon regeneration and circuit repair.

(c) So far, guidance modalities in adult have been studied at the injury site, whether in the optic nerve injury model or in the spinal cord injury model (Fu et al., 2010; Silver and Miller, 2004). In fact, repulsive guidance molecules have been shown to be upregulated at the injury site, leading to a broad growth-inhibitory effect (« chemical barrier ») rather than an actual guidance pattern. In our present work, we base our analysis on the assumption that the modulation of intrinsic growth properties has unlocked axon regeneration after injury (He and Jin, 2016). Having overcome the challenge of growth distance, the next step is to reroute regenerating axons onto the correct path to achieve functional repair. Hence, the novelty of our approach lies in the characterization of protein changes at distance of the injury site – which, as pointed by the Reviewer, have been already reported.

In our revised manuscript, we now provide the phenotype of receptor-silencing in both wild-type and long-distance regeneration conditions (updated Supplementary Figure 7 and updated Figure 8). Consistent with the result mentioned by the Reviewer and to work of other (Duffy et al., 2012; Joly et al., 2014), we found that in wild-type conditions, modulation of the Ephrin-B guidance signaling alleviates the growth barrier at the injury site, resulting in more axon regeneration. In a long-distance regeneration model, we find that modulation of Ephrin-B guidance signaling significantly increases the number of axons entering the chiasm and crossing the midline. Same is observed when modulating Sema4D signaling. Together, our in vivo experiments provide the compelling evidence that regenerating axons can be guided in the mature system. This result will be of primary importance in the field of axon regeneration and circuit repair.

Finally, the specificity of the effect is not documented here. In short, this preliminary result does not add substantially to the paper.

Regarding the specificity of our ex vivo proof-of-concept experiment, we now provide results for other candidate molecules, in addition to Ephrin-B3: Sema4D, which is also repulsive for adult axons, and Sema7A, which is neutral (updated Figure 7). For both these other candidates, we show expression of the corresponding receptors in RGC. We moved forward with the in vivo experiments for the repulsive cues Ephrin-B3 and Sema4D, as commented above and described in our manuscript.

Reviewer #3 (Remarks to the Author):

To address the yet unsolved problem of insufficient re-innervation and functional recovery of nerves in the central nervous system after injury, Vilallongue and colleagues generated and analyzed a rich proteomic data set for regenerative axons. The work is following a clear rationale and is definitely technically sound. The authors analyzed the proteomes of four different regions in the murine brain, which are targeted by retinal ganglion cell axons. Interestingly, these regions show clear changes in their proteomes after bilateral optic nerve injury. By functional ex-vivo assays, the work shows that RGC axons are still responsive to guidance cues. The proteomically defined guidance landscape provided by the authors is supporting the hypothesis that inefficient re-innervation is caused by miss-guidance rather than a guidance block. Thus, the data set clearly provides novel insights into the mechanisms of neurite regrowth in response to an injury and my serve as a sound data foundation for future functional studies. The conclusions drawn are convincing as they are clearly justified by the experimental data provided. The data are documented according to the standards of the field satisfying the MIAPE (Minimum Information About a Proteomics Experiment) standards for quantitative proteomic data.

We thank the Reviewer for his/her positive comments on our work.

I just have minor comments:

(1) Page 24/ line 530: As recommended by the MIAPE standards, the number of entries (Mus musculus taxonomy) should be given along with the version of the database used for protein/ peptide identification.

We proceeded according to the Reviewer's suggestions. The Material and Methods were modified to add this information, as followed:

"Data were processed automatically using Mascot Distiller software (version 2.7.1.0, Matrix Science). Peptides and proteins were identified using Mascot (version 2.6) through concomitant searches against Uniprot (Mus Musculus taxonomy, for Optic chiasm and SCN samples: downloaded in June 2019 with 87573 entries; and for dLGN, vLGN and SCol samples: downloaded in August 2020 with 87975 entries)" (lines 612-616).

(2) The raw data have been uploaded to PRIDE. I would recommend to provide an experimental template allowing to assign single runs to the different conditions. In addition, it would be good to provide a more detailed description of the experimental parameters there.

We proceeded according to the Reviewer's suggestions. We now provide a detailed description for Sample Processing Protocol and Data Processing Protocol with the data deposited on PRIDE. We also provide an experimental design excel file to explain which raw file corresponds to which tissue, condition and replicate.

(3) The figures are of sufficient quality, comprehensibly summarizing the outcome of the data analysis. Nevertheless, the fonts are very small at a quite low resolution making it sometimes difficult to read the labels. The authors might consider to increase the resolution and/or font size.

We proceeded according to the Reviewer's suggestions. We have updated accordingly Figures 2, 3, 4 and 6, and Supplementary Figures 2, 4 and 6 for better reading.

In conclusion, given its significance for the field as well its sound technical quality, I can recommend the manuscript for its publication in Nature Communications with minor revisions.

REVIEWERS' COMMENTS

Reviewer #1 (Remarks to the Author):

The authors have resolved all my concerns, but the manuscript would still benefit from a clearer acknowledgment of what was known from prior studies about the roles of the specific guidance cues/receptors, which were further investigated in the present study, and clearly explaining what new knowledge about these guidance cues/receptors was added by the present study. Also, it is important to discuss whether the observed effect on axon regeneration by these guidance cues/receptors may not recapitulate the developmental path, and, possibly, rather misroute the axons towards wrong targets, which ultimately may not synapse with the incoming axonal terminals that would not have the matching cues/receptors; and even if they would synapse, the information carried by these axons may not be compatible with the targets' efferent information-processing pathways.

Reviewer #2 (Remarks to the Author):

The authors have responded fully to my criticisms of the initial submission.

Reviewer #1 (Remarks to the Author):

The authors have resolved all my concerns, but the manuscript would still benefit from a clearer acknowledgment of what was known from prior studies about the roles of the specific guidance cues/receptors, which were further investigated in the present study, and clearly explaining what new knowledge about these guidance cues/receptors was added by the present study.

We thank the Reviewer for his/her positive comment on our revised manuscript. We had described the known effects of these guidance signals in the results part (lines 325-334). In this revised version, we describe as well their known implication in the developing and mature nervous system in the discussion (lines 518-526). So far, they have been described as inhibitory of axon growth upon injury. We stress out that the novelty of our study is to address their function as guidance cues in the mature regenerating nervous system (lines 518-526)

Also, it is important to discuss whether the observed effect on axon regeneration by these guidance cues/receptors may not recapitulate the developmental path, and, possibly, rather misroute the axons towards wrong targets, which ultimately may not synapse with the incoming axonal terminals that would not have the matching cues/receptors; and even if they would synapse, the information carried by these axons may not be compatible with the targets' efferent information-processing pathways.

We totally agree with the Reviewer that this is indeed a key point. We added it in the discussion (lines 530-533). Our study is pioneering axon guidance in the mature system and we hope it will open the way to bring more interest in this critical process. There are still many unknowns that are beyond the scope of this work.